# Weak acids produced during anaerobic respiration suppress both photosynthesis and aerobic respiration

Xiaojie Pang [1,2], Wojciech J. Nawrocki [3,5], Pierre Cardol [4], Mengyuan Zheng [1,2], Jingjing Jiang[1], Yuan Fang[1,2], Wenqiang Yang[1,2], Roberta Croce [3] & Lijin Tian [1,2] ✉

While photosynthesis transforms sunlight energy into sugar, aerobic and anaerobic respiration (fermentation) catabolizes sugars to fuel cellular activities. These processes take place within one cell across several compartments, however it remains largely unexplored how they interact with one another. Here we report that the weak acids produced during fermentation down-regulate both photosynthesis and aerobic respiration. This effect is mechanistically explained with an "ion trapping" model, in which the lipid bilayer selectively traps protons that effectively acidify subcellular compartments with smaller buffer capacities – such as the thylakoid lumen. Physiologically, we propose that under certain conditions, e.g., dim light at dawn, tuning down the photosynthetic light reaction could mitigate the pressure on its electron transport chains, while suppression of respiration could accelerate the net oxygen evolution, thus speeding up the recovery from hypoxia. Since we show that this effect is conserved across photosynthetic phyla, these results indicate that fermentation metabolites exert widespread feedback control over photosynthesis and aerobic respiration. This likely allows algae to better cope with changing environmental conditions.

Photoautotrophic species rely on both photosynthesis and respiration. While the former process harvests sunlight and stores its energy in the form of sugars, aerobic and anaerobic respiration (fermentation) oxidizes sugars releasing energy to meet metabolic requirements[1,2]. For *Chlamydomonas reinhardtii* (*Chlamydomonas* throughout), a model soil-dwelling alga, fermentation has been demonstrated to be important for survival in the weak light environment in the morning and evening[3–5], and it has been proven to be the preferred metabolic process at night, even when oxygen is not limiting[6]. In *Chlamydomonas*, photosynthesis and aerobic respiration occur in chloroplasts and

mitochondria, respectively[7], while fermentation pathways could independently occur in the cytoplasm, mitochondria or chloroplasts[8]. The activity of the enzymes participating in the photosynthetic and respiratory metabolism is finely regulated and often separated not only in space but also in time[7,9]. Nonetheless, these metabolic processes are inevitably intertwined since their products can passively- and actively cross the cellular membranes[10].

Additionally, a tight energetic coupling was found between chloroplasts and mitochondria in *Chlamydomonas* via a redistribution of reducing power[11,12]. In phylogenetically distant microalgae such as

[1]Key Laboratory of Photobiology, Institute of Botany, Chinese Academy of Sciences, 100093 Beijing, China. [2]University of Chinese Academy of Sciences, 100049 Beijing, China. [3]Department of Physics and Astronomy and LaserLab Amsterdam Faculty of Science, Vrije Universiteit Amsterdam, 1081 HV Amsterdam, The Netherlands. [4]Génétique et Physiologie des Microalgues, InBioS/Phytosystems, Institut de Botanique, Université de Liège, B22, 4000 Liège, Belgium. [5]Present address: Laboratoire de Biologie du Chloroplaste et Perception de la Lumière chez les Microalgues, UMR7141, Centre National de la Recherche Scientifique, Sorbonne Université, Institut de Biologie Physico-Chimique, 13 Rue Pierre et Marie Curie, 75005 Paris, France. ✉e-mail: ltian@ibcas.ac.cn

diatoms or Euglenas, metabolite exchange between aerobic pathways has also been shown[13,14].

Under anaerobic conditions, cells produce various weak acids, such as pyruvate, lactate, formic and acetic acid[15]. Under certain stress conditions, the concentration of those acids builds up in the medium to mM-range within an hour of hypoxia[8]. Currently, how these metabolites affect photosynthesis is poorly understood. This can be relevant because it has been shown that the addition of weak acids into a suspension of *Chlamydomonas reinhardtii* cell induces immediate, reversible non-photochemical quenching (NPQ) of chlorophyll fluorescence and a slower reduction of the plastoquinone (PQ) pool[16]. These effects indicate that acetic acid permeates through 4 membranes to acidify the thylakoid lumen and induce protonation of LHCSR3 and LHCSR1, the proteins responsible for low pH-sensing in the alga[17–21]. Thus, weak acids and potentially their salts could also impact photosynthesis. Interestingly, upon long-term dark adaptation, thylakoid lumen acidification was often observed in unicellular algae (hereafter referred to as dark-induced acidification)[22,23] and it was proposed to depend on either chlororespiration or ATP hydrolysis in darkness[22,24–26]. However, considering that long-term dark adaptation of highly concentrated micro algae often leads to anoxia, and thus fermentation, which converts sugar into acids[8], we speculate that this dark-induced lumen acidification is due to fermentation.

To verify this hypothesis and to explore the interplay between metabolism and photosynthesis, we examined photosynthetic activity by monitoring the chlorophyll fluorescence of *Chlamydomonas* during fermentation. We show that the weak acids produced by fermentation acidify the thylakoid lumen. We demonstrate that the metabolic control of the chloroplast by fermentation is passive and does not depend on ATP synthase or chlororespiration, and we propose a mechanism for the metabolic control induced by fermentation. Finally, we show that the weak acids not only strongly decrease photosynthetic light harvesting by promoting NPQ, but also decrease aerobic oxygen consumption. These side effects of fermentation represent a previously unknown case of feedback control by metabolites regulating aerobic respiration and the onset of photosynthesis.

## Results

### Fermentation acidifies the lumen in *Chlamydomonas* due to weak acid production

Fermentative metabolism in *Chlamydomonas* generates various weak acids[8]. To investigate how these acids possibly affect photosynthesis, we monitored chlorophyll fluorescence during fermentation, focusing on the quenching induced by lumen acidification. To separate fluorescence quenching from state transitions (induced in darkness in anoxic conditions[27,28]), the *stt7-9* mutant, deficient in the Stt7 kinase required for this antenna redistribution process[29], was used. Figure 1a shows significant fluorescence quenching during anaerobic respiration. The quenching relaxed completely after the addition of KOH (after fermentation, the cell culture becomes acidic, and KOH is used to adjust the pH of the medium back to 7.5). Similar results were obtained for WT CC-124 cells, even though the maximum fluorescence before and after quenching was not the same because of the presence of state transitions (Supplementary Fig. 1). The dependency of the quenching on LHCSR3 was shown using the double mutant *npq4 stt7-9* (see Fig. 1b, in which Fm' of *npq4 stt7-9* was stable through the fermentation process). As the LHCSR-dependent chlorophyll fluorescence can be used as a reliable lumenal pH indicator[30], we concluded that fermentation acidifies the lumen in *Chlamydomonas*.

To verify if quenching is caused by the accumulation of weak acids produced during fermentation, we ran the same experiments on *Chlamydomonas eustigma* NIES-2499, a species that lacks the enzymes involved in organic acid fermentation pathways while retaining the alcohol fermentation pathways[31]. Note that these cells perform normal acid-induced NPQ (Fig. 1c). During fermentation, the fluorescence of *C.*

*eustigma* cells decreased to some extent, but the change was not sensitive to KOH, indicating that it is not due to NPQ, but likely to state I → state II transition (Fig. 1d).

To further prove that the quenching of fluorescence during fermentation was indeed due to weak acids, we measured the extracellular weak acid content of *Chlamydomonas* at different time points during anoxia (every 30 min) and correlated them with NPQ values. The results show a positive correlation between formic and acetic acid concentrations and NPQ (Fig. 1e, f and Supplementary Fig. 2), in agreement with previous quantifications with externally-added acid[17]. In contrast, in the acidophilic algae NIES-2499, the cells hardly produced any acids (Supplementary Fig. 2), in agreement with previous measurements[31], thus the fluorescence is not quenched under hypoxia (Fig. 1d). Altogether, these results indicate that lumen acidification in the dark is due to organic acid fermentation and not to alcoholic fermentation.

### Exogenously added weak acids and their salts acidify the lumen in *Chlamydomonas*

Intriguingly, as reported earlier[17], many different weak acids other than acetic acid induce lumen acidification when added to the cell solution (a list of weak acids inducing this effect is shown in Supplementary Table 1 and Supplementary Fig. 3). We also exclude an effect of acetate incorporation in the cellular metabolism as the trigger of fluorescence quenching, as in the *icl* mutant, deficient in isocitrate lyase[32], the quenching phenotype remained (Supplementary Fig. 4). Moreover, not only acetic acid (Fig. 2a), but also 50 mM sodium acetate (NaAc) was able to induce quenching (Fig. 2b). The fluorescence quenching was fully relaxed upon addition of 100 μM nigericin, a known proton antiporter[33], indicating that NaAc induced lumen acidification, even though the pH of the solution remained unaffected. On the contrary, strong acids and their salts, like HCl and NaCl, did not induce quenching (Fig. 2c, d).

Above we showed that lumen acidification occurs in response to anoxic metabolism, and highlighted that the same effect was obtained by the exogenous addition of weak acids and their salts. As the use of weak acid considerably speeds up the experiments (seconds vs. hours[16,17]), we used this approach to test the presence of this phenomenon in a range of organisms across the photosynthetic phyla, including another green alga (*Chlorella pyrenoidosa* FACHB-9), a red alga (*Porphyridium purpureum* FACHB-840), a diatom (*Phaeodactylum tricornutum* FACHB-863), a moss (*Physcomitrella patens*) and several angiosperms (*Arabidopsis thaliana, Triticum* sp*., and Sorghum* sp.). The results in Fig. 3 and Supplementary Fig. 5a, c and d demonstrate that fluorescence quenching could be induced by adding acetic acid in all organisms. In the species for which NPQ mutants were available, we were able to confirm that the quenching was related to lumen acidification and dependent on pH-sensing proteins i.e., LHCSR in *Chlamydomonas* (Supplementary Fig. 6) and PsbS in Arabidopsis (Fig. 3e and Supplementary Fig. 5b).

### Is lumen acidification an active bioenergetic process or a passive chemical effect?

To understand the mechanism of weak acid-induced quenching, we sought to distinguish whether an active bioenergetic/metabolic pathway or a passive chemical effect is responsible for the observed effect. It was previously suggested that chlororespiration, a chloroplastic respiratory electron transfer chain involving the enzymes, PTOX2[34] and NDA2[35], could be responsible for lumen acidification in the presence of a reduced carbon source[16]. To test this hypothesis, we used the two chlororespiratory mutants *ptox2* and *nda2* and their parental strain WT CC-4533. We found that in all of them the addition of 50 mM NaAc could successfully induce fluorescence quenching (Supplementary Fig. 7 and Fig. 4a–c). Since the use of NaAc allowed keeping the pH of the cell medium neutral, and the quenching

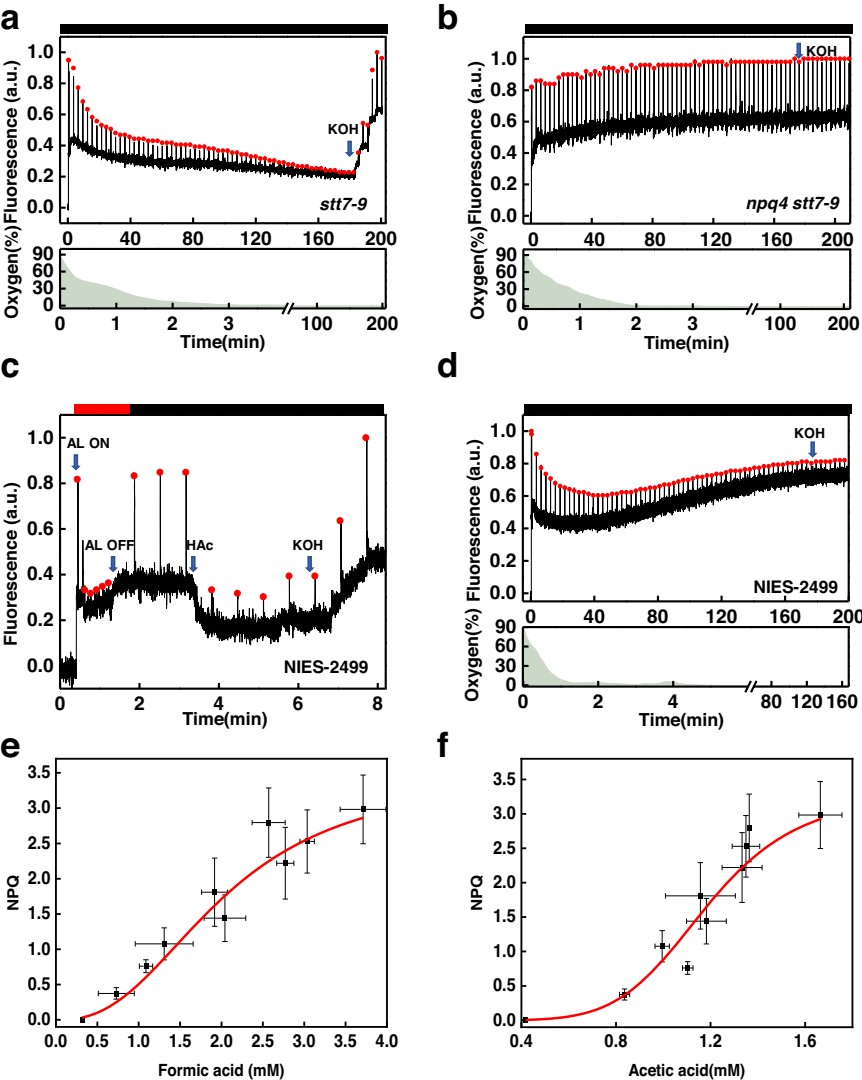

**Fig. 1 | Probing lumen acidification using NPQ during anaerobic treatment.**
Chlorophyll fluorescence and oxygen concentration were simultaneously recorded on *Chlamydomonas* mutants *stt7-9* (**a**), *npq4 stt7-9* (**b**), and on *Chlamydomonas eustigma* NIES-2499 in aerobic condition (**c**), and NIES-2499 in anaerobic condition (**d**). The cells used for the experiments shown in (**a**, **b**, **d**) were grown under high light and resuspended in fresh HSM-ficoll (10%) (**a**, **b**) or M-Allen -ficoll (**d**) in a cuvette and sealed to induce anoxia. The ficoll is used to keep the cells in suspension. Fm', indicated by the red dots, was recorded every 3 min. The addition of KOH completely relaxed the quenching in the *stt7-9*, but hardly affected the fluorescence in the double mutant *npq4 stt7-9* and NIES-2499. For the experiment shown in (**c**), cells were first exposed to actinic light (AL 1500 μmol photons m$^{-2}$ s$^{-1}$)

to induce chlorophyll quenching (20 s–80 s) in aerobic environment. After NPQ was fully released (80 s–200 s), a second round of fluorescence quenching was induced by adding 4.5 mM HAc (the pH of medium ~3.3), and after that, the addition of KOH (the pH of medium ~5.5) fully abolished the fluorescence quenching. The red bars represent actinic light illumination and the black bars dark treatment. The relationship between fermentation products formic acid (**e**) and acetic acid (**f**) accumulated in the medium of *Chlamydomonas* cultures and NPQ in *stt7-9*. Cells were kept in anaerobiosis in the dark for up to 4.5 h. Samples were taken at the indicated time points (0, 0.5, 1, 1.5, 2, 2.5, 3, 4 and 4.5 h), centrifuged, and filtered, and the medium was analyzed by HPLC. Data are taken from triplicate samples derived from three independent experiments. Error bars represent SD.

relaxed following the addition of nigericin, we conclude that chlorespiration is not a prerequisite for the salt-induced lumen acidification.

Another mechanism that could account for lumen acidification in the dark is the hydrolysis of ATP. When glycolysis and respiration are active, ATP produced in the process could be imported into the chloroplast and converted by the ATP synthase into a proton gradient[24,36]. To verify whether the lumen acidification was ATP synthase-dependent, we used the *FUD50* mutant, which bears a deletion in the atpB gene[37]. To exclude Fm changes due to state transition, state I was induced by weak light pre-illumination in the presence of DCMU[38]. The acid-induced quenching was also present in *FUD50* (Fig. 4d), showing that lumen acidification is not caused by ATP hydrolysis. Quenching induced by NaAc was fully relaxed by adding nigericin,

indicating that a buildup of ΔpH took place across the thylakoid membrane also in this mutant.

## The lumen acts as an "ion trap"
What is the mechanism of lumen acidification? To answer this question, we spatially mapped the pH distribution in various intracellular compartments using the pH-sensitive dye BCECF. Cell wall-free mutant CC-400 was used to ensure easier penetration of the dye into the cell. Confocal fluorescence images showed that the dye was uniformly distributed throughout the cytoplasm but did not enter the chloroplast (Fig. 5a and Supplementary Fig. 8). To accurately determine cytoplasmic pH, a calibration curve was generated (Fig. 5b). The results show that the addition of 50 mM NaAc hardly changes the cytoplasmic pH. Conversely, the addition of HAc brings the cytoplasm pH below 5.5

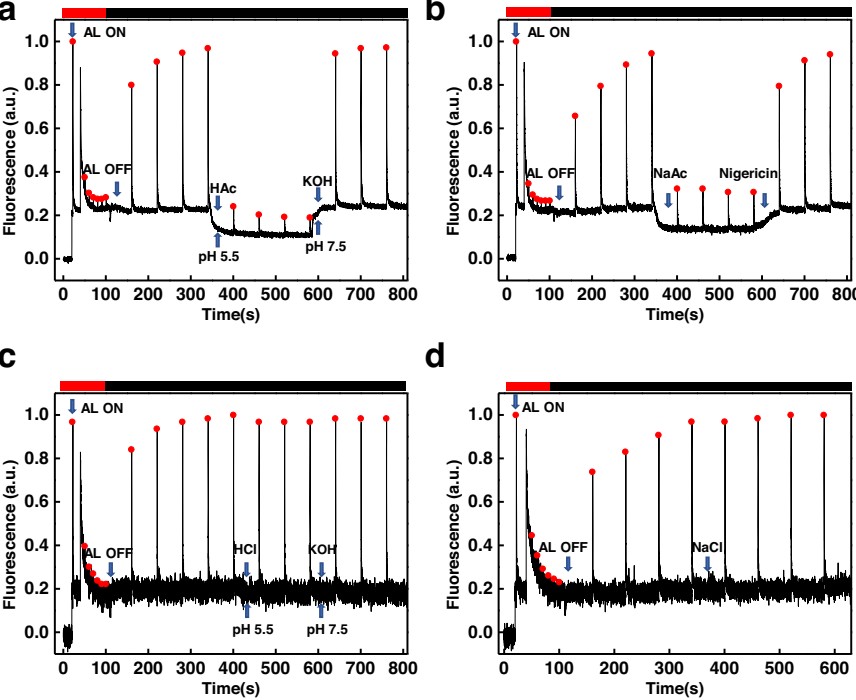

**Fig. 2 | Typical chlorophyll fluorescence traces of *stt7-9* cells measured with Dual-PAM.** Cells were illuminated with a strong actinic light (AL, 1500 µmol photons m$^{-2}$ s$^{-1}$) for 150 s followed by a dark recovery (4 min). **a** The addition of 4.5 mM HAc (the pH of medium 5.5) caused fluorescence quenching and the addition of KOH (the pH of medium 7.5) abolished it. **b** The addition of 50 mM NaAc leads to fluorescence quenching (350 s–650 s) which was abolished upon addition of 100 µM nigericin (650 s–810 s). **c** Adding HCl (pH of medium 5.5), and KOH to titrate the pH back to 7.5 did not affect fluorescence. **d** The fluorescence did not change when the 50 mM NaCl was added.

(Fig. 5b). In either case, the cells were not significantly shrinking in size, excluding osmotic stress as a cause of lumen acidification (see the confocal images Supplementary Figs. 9 and 10). Finally, we showed that the cytoplasmic pH upon anaerobic respiration was <5.5 (see Supplementary Table 2).

According to the partition theory, ionized molecules usually cannot penetrate, cell membranes because they are hydrophilic and poorly lipid-soluble[39]. That explains why strong acid and strong acid salts fail to lead to fluorescence quenching (given they are virtually fully dissociated in solution), but weak acids and their salts could induce fluorescence decrease. Based on these results, we propose that the chloroplast lumen acts as an "ion trap" (Fig. 5c). For acids, an equilibrium between their protonated and ion form is described by the Henderson-Hasselbalch equation: $pH = pK_a + \log_{10}\left([A^-]/[HA]\right)$. A key feature of a membrane is that it allows small uncharged molecules to diffuse through. Therefore, the non-dissociated form of weak acids can penetrate the cell membrane all the way to the thylakoid lumen.

To understand why there is a difference in pH between the thylakoid lumen and other compartments, their buffering capacity needs to be considered. It has been reported that the buffer capacity of the lumen (0.8–1.0 mM) is much lower than that of cytoplasm (~20 mM) and stroma ($27 \pm 4$ mM)[40–42]. Additionally, the total volume of the lumen is very small compared to the stroma[43]. When the concentration of HAc/NaAc increases in the cytoplasm, the concentration of its protonated form, HAc, also increases. These charge-free molecules diffuse across the membranes and then release protons in the lumen, consequently acidifying it as its buffer capacity is weak. An estimate of the pH values of two compartments separated by a membrane and with different buffer capacities is reported in Supplementary Table 3.

### Effect of weak acids on photosynthesis

To understand the physiological relevance of the observed acidification, we evaluated the effect of weak acids on photosynthetic electron transfer. To exclude effects from NPQ and state transitions, which would further negatively feedback the electron transfer rate (ETR) upon acid addition, we used the *npq4 stt7-9* double mutant. The results showed that the relative electron transport rate of both PSII (ETR(II)) and PSI (ETR(I)) dropped significantly in acidic conditions (Fig. 6a, b). We also examined the effect of weak acids produced by anaerobic fermentation showing that ETR(I) in *npq4 stt7-9* dropped in anaerobic conditions (Supplementary Fig. 11a, b). These results indicate that electron transfer is indeed hampered by the low lumenal pH[44,45]. The effect can be due to the non-photochemical reduction of the PQ pool, and/or a decrease in the activity of cytochrome $b_6f$, which is known to be down-regulated by the low lumenal pH[46].

To disentangle these effects, we measured the electrochromic shift signal (ECS)[47]. The b phase of the ECS is related to the cytochrome $b_6f$-mediated electron transfer, and it is expected to slow down upon lumen acidification[48]. The latter slows down quinol oxidation at the Q$_o$ site due to the initial deprotonation event being an uphill reaction with a higher energetic threshold at low luminal pH. The b-phase in the ECS signal (ms-timescale) was slower in anaerobiosis (Supplementary Fig. 11c). We also confirmed the absence of the b-phase in the ECS signal (ms-timescale) of the *npq4 stt7-9* under aerobic conditions when 4.5 mM HAc or 50 mM NaAc was added (Supplementary Fig. 12).

In summary, the data show that weak acids, which are produced by fermentation, can suppress both light harvesting−by triggering NPQ−and ETR. To understand if this effect has a physiological role, we measured the oxygen levels during photosynthetic re-activation of the anaerobic-acclimated cells. The results showed that in low light oxygen evolution was quickly restored in 20 min when the light was turned on after 3 h of anoxia (Fig. 6c). In contrast, when KOH (pH of the medium 7.5) was added to neutralize acidification, oxygen was released slowly with a time delay of 1 h compared to the sample without KOH (Fig. 6c). To exclude a toxic effect of KOH, cells were also measured in aerobic conditions. No difference in oxygen evolution was observed for cells

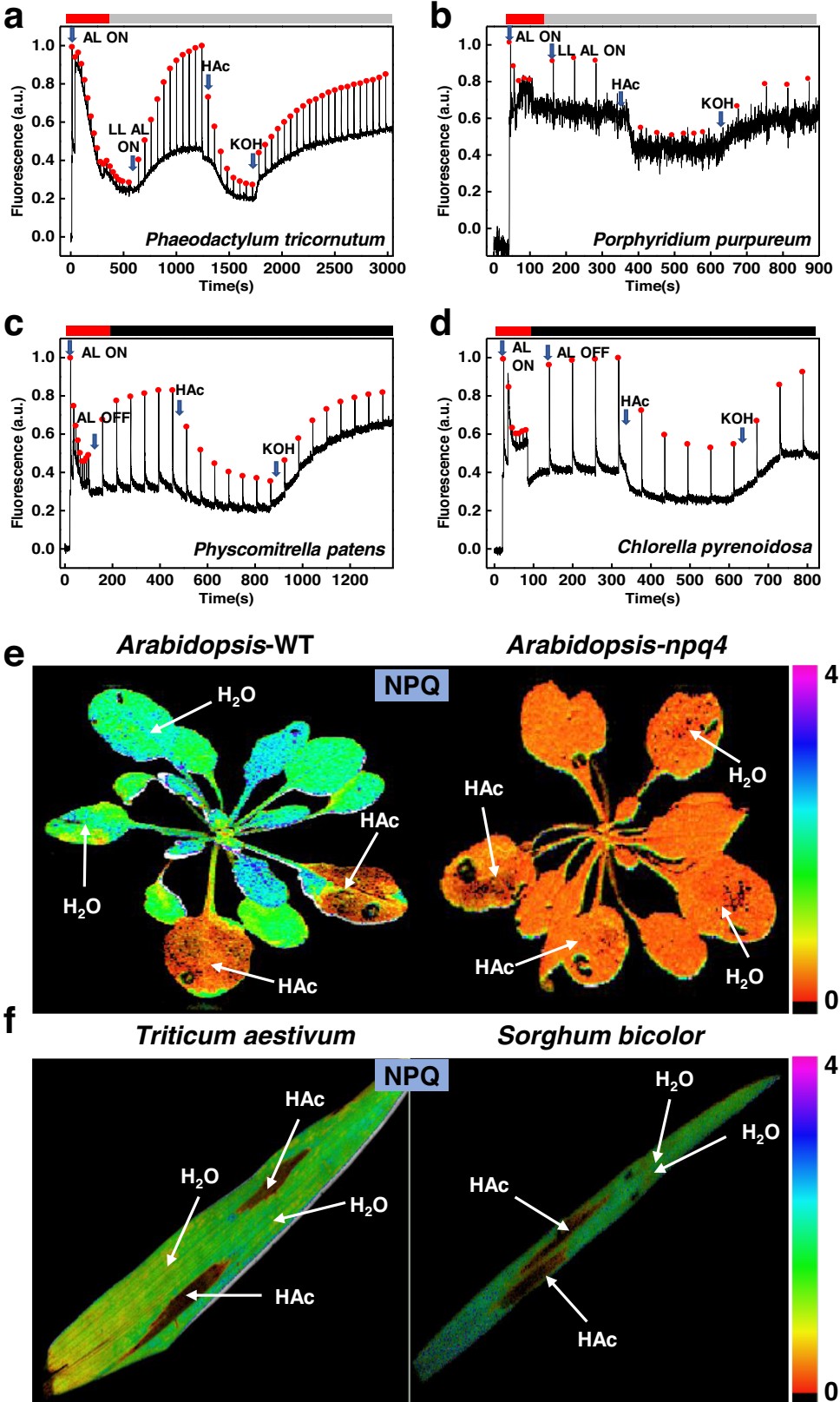

with and without KOH (Supplementary Fig. 13). We also measured the quantum yield of PSII Y(II) in cells in anaerobic conditions and found that the value was higher in the presence of KOH (Supplementary Fig. 14), indicating that acidification decreased the photosynthetic activity. Why then the addition of KOH slows down oxygen evolution during the onset of photosynthesis following anoxia? A possible explanation is that aerobic respiration is also inhibited by the weak acids produced during fermentation. Upon addition of KOH, aerobic respiration rapidly recovered. Photosynthesis also recovered, but in the first phase, at dawn, which is characterized by low light intensity, oxygen production is minimal. In these conditions the oxygen consumption rate exceeds the production rate, leading to a significant

**Fig. 3 | Chlorophyll fluorescence traces of algae and images of higher plants. a** A typical chlorophyll fluorescence trace of *Phaeodactylum tricornutum* (FACHB-863). NPQ was induced either with high light illumination (AL, 50 s–510 s), or by adding HAc in the dark (the pH of the medium was 5.5) at 1400–2000 s. The fluorescence quenching was abolished upon addition of KOH in weak actinic light (10 μmol photons m$^{-2}$ s$^{-1}$). **b** A typical chlorophyll fluorescence trace of the *Porphyridium purpureum* (FACHB-840). NPQ was first induced with high light illumination (50 s–180 s), then a second round of quenching was induced by adding HAc (pH of the medium: 5.5) at 400 s–650 s after full recovery of fluorescence. The second round of fluorescence quenching was recovered by adding KOH (the pH of the medium was 7.5) in weak actinic light (10 μmol photons m$^{-2}$ s$^{-1}$). **c** Chlorophyll fluorescence trace of *Physcomitrella patens*. Adding HAc (pH of the medium: 5.5) caused chlorophyll fluorescence quenching and the addition of KOH (the pH of

medium 7.5) abolished the quenching. **d** Similar chlorophyll fluorescence trace of *Chlorella pyrenoidosa* (FACHB-9) as obtained in (**c**). The addition of HAc (pH of medium 5.5) caused chlorophyll fluorescence quenching and the addition of the KOH (titration of the medium pH back to 7.5) deactivated the quenching. **e** The NPQ images of wild-type *Arabidopsis* Col-0 (left) and *npq4* mutant (right) after stable NPQ (250 s) was reached under actinic light (AL 400 μmol photons m$^{-2}$ s$^{-1}$). For each plant, two leaves were filtrated with 0.1 M acetic acid, while another two leaves were infiltrated with an equal amount of H$_2$O as control. **f** The npq images of *Triticum aestivum* (left) and *Sorghum bicolor* (right) after a stable NPQ was reached under actinic light (400 μmol photons m$^{-2}$ s$^{-1}$). For each leave, two parts of a leaf were infiltrated with 0.1 M acetic acid, while another two parts were infiltrated with an equal amount of H$_2$O as control. All experiments were performed in triplicate.

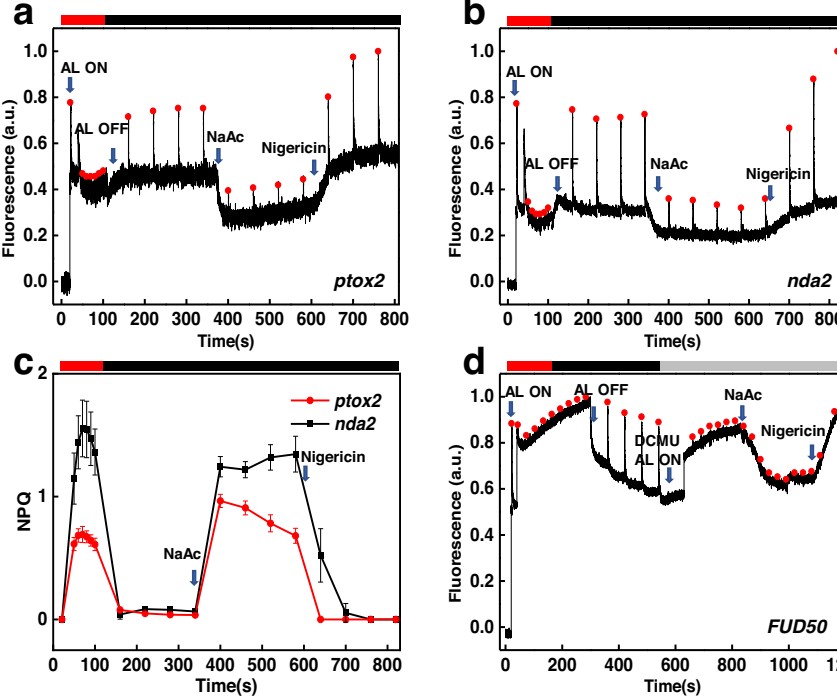

**Fig. 4 | Typical chlorophyll fluorescence traces of *ptox2*, *nda2* and *FUD50* mutants measured with Dual-PAM.** *ptox2* (**a**) and *nda2* (**b**) cells were illuminated with strong actinic light (1500 μmol photons m$^{-2}$ s$^{-1}$) for 150 s followed by dark recovery (4 min). The addition of 50 mM NaAc leads to fluorescence quenching (350 s–590 s) and nigericin cancels the quenching (650 s–850 s). **c** NPQ of *ptox2* and *nda2* calculated from the data presented in (**a**) and (**b**). The NPQ values represent an average of three independent measurements. Error bar, SD (*n* = 3). **d** *FUD50* cells were initially illuminated with strong actinic light (40 s–350 s)

followed by dark recovery (350 s–650 s). Then the cells were induced in state I (100 μM DCMU and weak actinic light 10 μmol photons m$^{-2}$ s$^{-1}$). The chlorophyll fluorescence quenching was induced with 50 mM NaAc (850 s–1000 s) and the quenching fully abolished upon addition of 100 μM nigericin (1000 s–1200 s). The red bars indicate strong actinic light illumination, the gray bar and black bars represent weak actinic light (10 μmol photons m$^{-2}$ s$^{-1}$) and dark treatment, respectively.

delay in oxygen accumulation. This hypothesis is supported by the fact that in high light, when oxygen production became the dominating process, the effect of KOH disappears (Fig. 6c). To further verify that aerobic respiration was reduced under acidification, we incubated the cells at different pH under aerobic conditions in the dark. The rate of oxygen consumption of the cells decreased under acidification conditions (Fig. 6d and Supplementary Fig. 15), demonstrating that aerobic respiration was severely suppressed under acidic conditions.

## Discussion

It has often been reported that lumen acidification occurs in green algae, diatoms, lichens and other species kept in darkness for hours[24,49–51]. This phenomenon was ascribed to either chlororespiration or ATP hydrolysis[52,53]. Indeed, if chlororespiration is electrogenic (i.e., the combined activity of NADPH dehydrogenase and plastoquinol terminal oxidase results in net proton translocation to the lumen), then

acidification takes place. Nevertheless, in the case of *Chlamydomonas*, the chlororespiratory enzymes PTOX and NDA2 are probably not electrogenic as they are both monotopic and chloroplast stroma-exposed[54]. Likewise, ATP synthase running in reverse mode could lower the lumen pH thanks to ATP hydrolysis[55] and it has been demonstrated that ATP (of mitochondrial origin) hydrolysis by chloroplastic ATP synthase occurs in dark oxic conditions in several species (Diatoms[13], Phaemonas[56], Euglens[14]), however, in anoxic conditions, this is unlikely to happen because mitochondrial respiration is abolished. As anticipated, using *ptox2*, *nda2* and *FUD50* mutants, we unambiguously demonstrated that the lumen acidification in the presence of weak acids does not originate from chlororespiration nor from ATP hydrolysis. Instead, we propose that lumen acidification is directly due to the weak acids produced during fermentation that translocate to the thylakoid lumen, which acts as an "ion trap": lipid membrane, impermeable to charged molecules, effectively traps

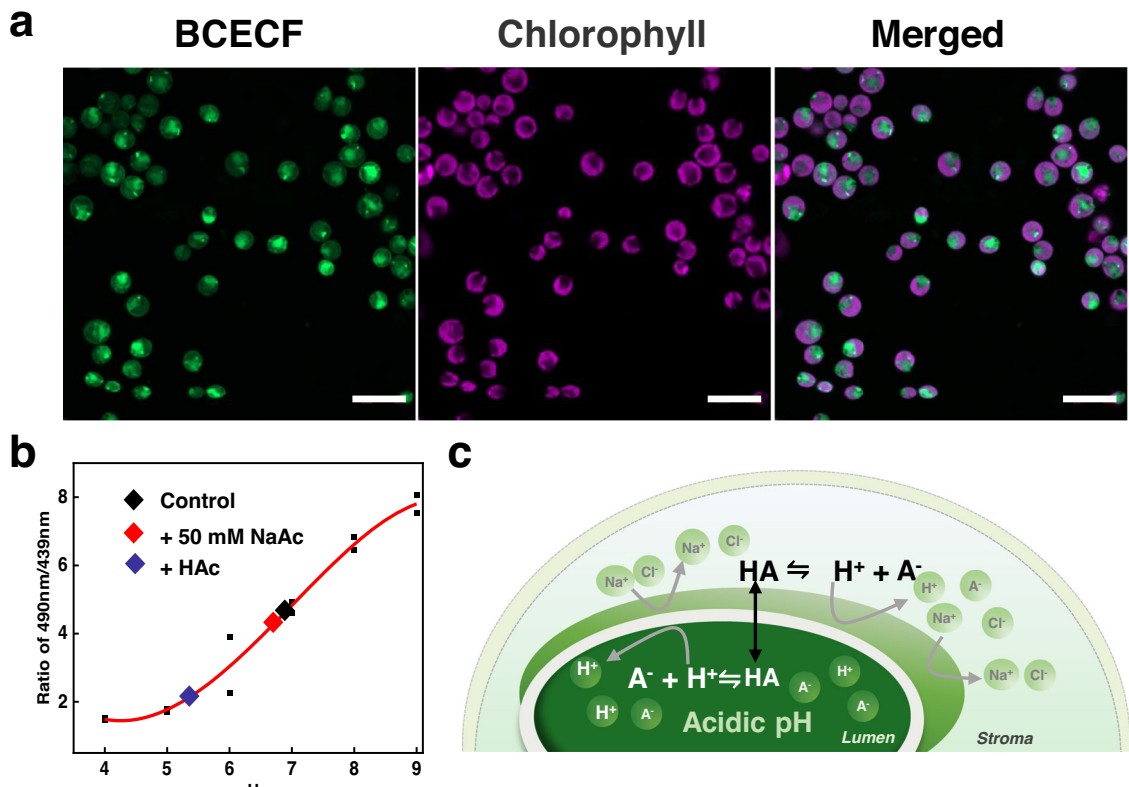

**Fig. 5 | Intracellular pH was measured with the pH-sensitive dye BCECF. a** Cells loaded with BCECF-AM using a confocal microscope (Leica TCS SP5). Fluorescence excitation was at 488 nm and emission at 515 ± 15 nm. Bar = 10 μm. Image of CC-400 cells, from left to right: BCECF fluorescence, chlorophyll fluorescence and the merged fluorescence. The experiment was performed in triplicate. **b** The intracellular pH calibration curve was measured by the excitation spectrum with emission at 530 nm. 50 mM NaAc (red diamond), 4.5 mM HAc (blue diamond) and untreated cells as a control (black diamond). The data points represent an average value from three independent measurements. The little black squares represent the averaged ratio of 490 nm to 439 nm at different pH values. **c** Model of "ion trap": the dark green area represents the thylakoid, and the outer semicircle is the chloroplast. HA stands for the acetic acid neutral form, while A⁻ for the acetate anion.

protons, which increasingly acidify the inner space with low buffer capacity. Hence, we report a new type of light-independent lumen acidification of cells in anoxia, in addition to the observed lumen acidification of cells in oxic condition led by ATP hydrolysis.

Additional evidence supports the "ion-trap" model. Recently, Gabba et al.[57] successfully determined the permeability coefficients of weak acids in lipid vesicles and living cells and they demonstrated that weak acids cross the membrane mainly via passive diffusion rather than protein-mediated transport. More importantly, in vitro experiments with the pyranine pH assay using liposomes with a low internal buffer capacity showed that the inside of the liposomes acidifies upon the addition of weak acid salts[57], in agreement with our results in vivo.

Understanding how fermentation, aerobic respiration and photosynthesis pathways are entangled requires a holistic view of the cell. As schematically illustrated in Fig. 7, we showed that the weak acids produced during fermentation could penetrate the thylakoid lumen thus suppressing the light reaction of photosynthesis. In parallel, we propose that they also penetrate the mitochondria slowing down aerobic respiration. The mitochondrial matrix has a low H⁺-buffering capacity (5 mM) compared with the cytosol (20 mM)[58] and in anaerobic conditions, it might be acidified by the weak acids produced by fermentation. This effect would collapse, even invert, the pH gradient across the membrane, which probably inhibits the electron transport chain and thus reduces the oxygen uptake. Yet, how weak acids suppress mitochondrial metabolism remains to be clarified.

In the natural environment, photosynthetic organisms experience an alternation of dark and low light conditions. Upon dusk, in photosynthetic microalgae, e.g., *Chlamydomonas*, oxygen is rapidly depleted by respiration and fermentation metabolism becomes active. Fermentation generates energy to maintain cell viability and supports cell redox balance by re-oxidizing NAD(P)H. Here we propose that along with these two functions, fermentation plays an additional role. The production of weak acids partially inhibits both photosynthesis and respiration, and the effective limitation of the later process can help the cell to quickly shift from anoxia to aerobic conditions when light intensity is low and respiration consumes more O₂ than photosynthesis produces. This process can be further enhanced in non-axenic conditions of microbial mats, with organisms exhibiting fermentative metabolism influencing the physiology of their neighboring cells. Why, then, would the light reactions of photosynthesis be suppressed? We suggest that the biological advantage of lowering the lumenal pH might still be photoprotective: weak acids produced by anaerobic fermentation lead to NPQ and a slower photosynthetic electron transport in the dark, which reduces electron input into the electron transport chain in conditions where electron sinks are limiting before the CBB cycle fully activate or alternative electron outlets, such as water to water cycle[59], chlororespiration[60], flavodiiron-or hydrogenase-dependent pathways[61] not yet/anymore active.

## Methods

### Algae and plants growth

In this study, *C. reinhardtii* wild-type CC-400[62], CC-124, CC-4533, the state transition mutant *stt7-9*[29], the double mutant *npq4 stt7-9* impaired in state transition and qE[38], the *ptox2* (LMJ.RY0402.152174), *nda2* (LMJ.RY0402.206160), *icl* and its wild-type CC-137[32] and *FUD50* (CC-1287) mutant were used, all of which are available from the

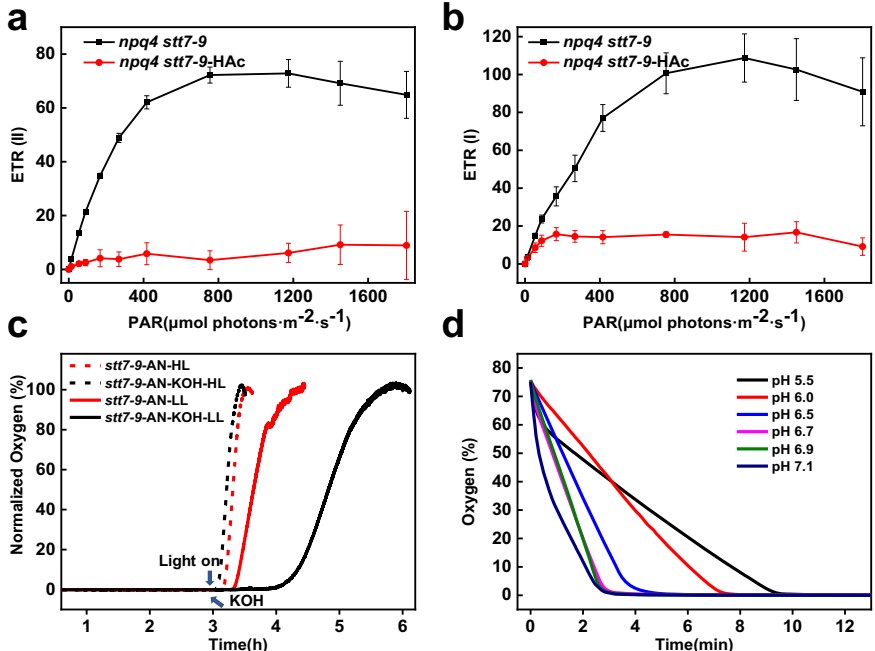

**Fig. 6 | Light-response curves of electron transport rates in *Chlamydomonas* and determination of oxygen levels in cell cultures. a** Light-response curve of relative electron transport rate of PSII, ETR(II) and **b** Light-response curve of relative electron transport rate of PSI, ETR(I) in *npq4 stt7-9* before and after adding HAc. Both measurements are performed on cells in aerobic conditions: control (black) and in the presence of 4.5 mM HAc (medium pH 5.5) (red), Error bar, SD (*n* = 4 biologically independent samples). **c** Oxygen level measured in *stt7-9* cells. Four samples with identical cell concentrations were kept under anaerobic conditions for 3 h before measurement. In two samples KOH was added (black arrow) and the cells were measured under moderate light illumination at 200 µmol photons m⁻² s⁻¹ (*stt7-9-AN-KOH-LL*, black solid) and under high light illumination at 600 µmol photons m⁻² s⁻¹(*stt7-9-AN-KOH-HL*, black dash). The other two samples were measured under the same two light conditions but in the absence of KOH. **d** Oxygen level measured in *stt7-9* cells at different pH under aerobic conditions. Six points were measured from pH 7.1 to 5.5, sequentially corresponding to concentration of acetic acid of 0, 0.5, 1.5, 2.3, 4.1, and 5.0 mM.

*Chlamydomonas* Resource Center in the University of Minnesota (http://www.chlamycollection.org/). The mutants were verified by PCR (Supplementary Fig. 16 and Supplementary Table 4). Each strain (except for the *FUD50* and *npq4 stt7-9*) was grown under high-light (350 µmol photons m⁻² s⁻¹) in high-salt medium (HSM) at 25 °C till to its log phase[63]. The *FUD50*, a deletion in the atpB gene of the chloroplast ATP synthase, was grown in Tris acetate-phosphate medium (TAP) at 25 °C under low light (20 µmol photons m⁻² s⁻¹) till to logarithmic phase and then they were exposed to high light (350 µmol photons m⁻² s⁻¹) for 3 h to induce LHCSR proteins expression. The *npq4 stt7-9* was grown under low light (20 µmol photons m⁻² s⁻¹) till to logarithmic phase.

*Chlamydomonas eustigma* NIES-2499 was purchased from the Microbial Culture Collection at the National Institute for Environmental Studies in Japan (NIES) on a rotary shaker in photoautotrophic medium M-Allen at pH 3.5[31], at 25 °C under high light (350 µmol photons m⁻² s⁻¹).

*P. tricornutum* (diatom) wild-type strain was grown in F/2 medium at 19 °C under light illumination of 40 µmol photons m⁻² s⁻¹. *Chlorella pyrenoidosa* (FACHB-9) was maintained in BG11 medium at 25 °C under light illumination of 350 µmol photons m⁻² s⁻¹. *Porphyridium purpureum* (red alga) was grown in artificial seawater (ASW) at 28 °C[64]. All algal cells were grown on a rotary shaker at 120 rpm in Erlenmeyer flasks. Cells were harvested from their exponential growth stage and resuspended in a fresh medium before measurements.

Wild type of *Physcomitrella patens* (moss) were maintained on minimum PpNO3 solidified with 0.75% plant agar at 25 °C[65], 16 h light/8 h dark under normal light (50 µmol photons m⁻² s⁻¹). For high light treatments, 6-days-old plants were transferred to 350 µmol photons m⁻² s⁻¹ for 2 h. *Triticum aestivum* (wheat) and *Sorghum bicolor* (Sorghum), *Arabidopsis thaliana* Col-0 and the PsbS-deficient mutant *npq4* (8 weeks old plants) were grown at normal light

(50 µmol photons m⁻² s⁻¹) at 21 °C with 12 h light/12 h dark photoperiod.

### Verification of the mutants *ptox2*, *nda2* and *FUD50*

PCR confirmation of mutants *ptox2* (LMJ.RY0402.152174) and *nda2* (LMJ.RY0402.206160) that originally purchased from the *Chlamydomonas* Resource Center: a paromomycin resistance gene cassette-CIB1 was inserted in the gene in the *ptox2* and *nda2* mutant, which was obtained from the *Chlamydomonas* Library Project (CLiP)[66,67]. We confirmed that the cassette CIB1 has an insertion in the intron of the PTOX2 gene in the *ptox2* mutant and an insertion in the 5′-UTR of the NDA2 gene in the *nda2* mutant by the PCR genotyping (Supplementary Fig. 16a, b). We performed semi-quantitative RT-PCR, and found no detectable expression of PTOX2 and NDA2 in the mutants *ptox2* and *nda2*, respectively (Supplementary Fig. 16e, f).

All the PCR fragments were also confirmed by DNA sequencing.

PCR confirmation of mutant *FUD50* (Supplementary Fig. 16c): this mutant lacks atpB gene, thus fails to assemble the ATP synthase. It was verified by primers FUD50-F/FUD50-R(refers to[68]).

### Fluorescence measurements with Dual-PAM 100/Image-PAM in anaerobic/aerobic conditions

Chlorophyll fluorescence traces were obtained with a pulse amplitude-modulated fluorimeter (Dual-PAM 100, Walz). Saturating light (6000 µmol photons m⁻² s⁻¹, 250 ms), red actinic light (AL, 1500 µmol photons m⁻² s⁻¹) and red measuring light were used through all measurements. The NPQ and Y(II) were determined by NPQ = (Fm − Fm′)/Fm′ and Y(II) = (Fm′ − F)/Fm′[69]. Fm is the maximum fluorescence for dark-adapted cells, Fm′ represents the max fluorescence under light, and F is the steady-state fluorescence in the light.

The effective quantum yield of PSI [Y(I)] was estimated from the absorbance changes of the dual-wavelength (820–870 nm) measured

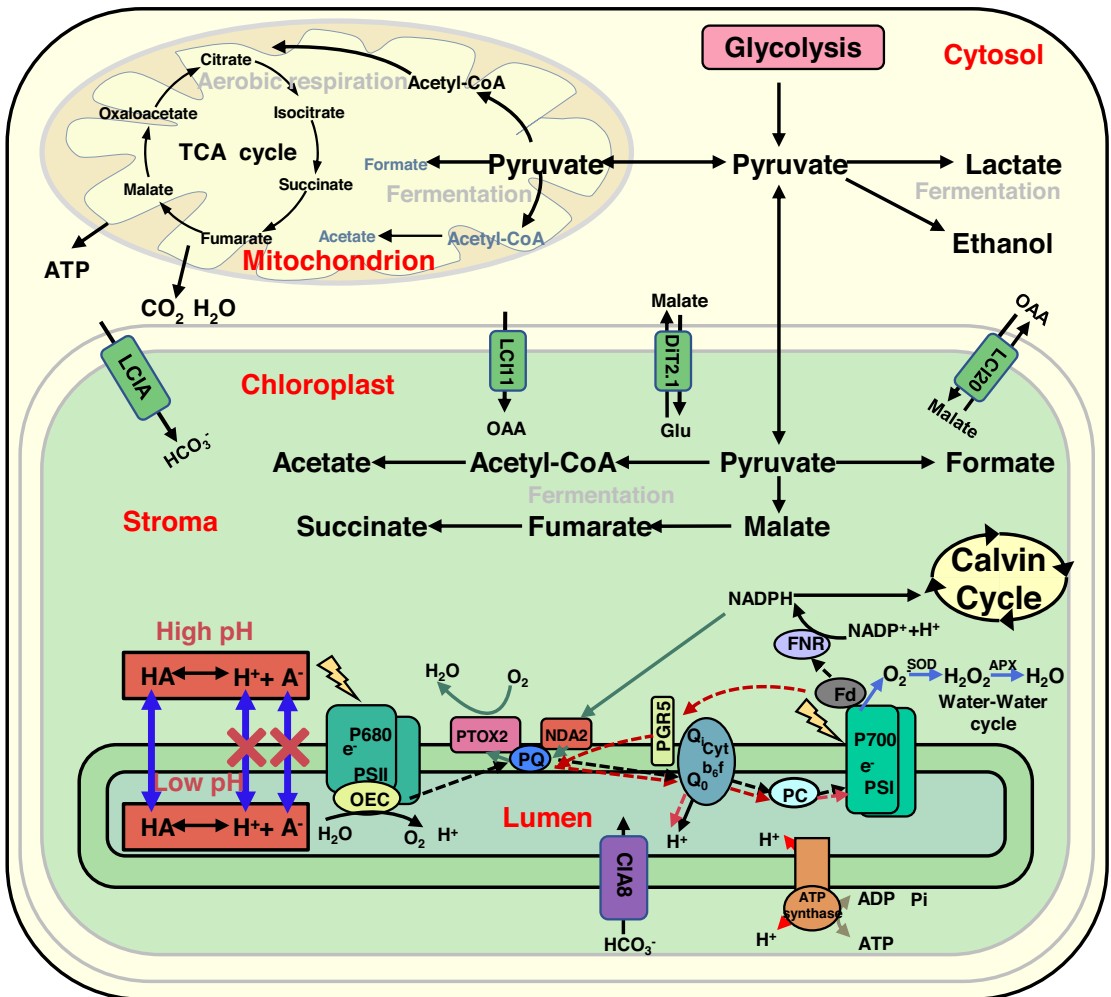

**Fig. 7 | Simplified scheme of metabolic pathways for *Chlamydomonas* under anoxic conditions.** The figure illustrates the possible interplays between fermentation, respiration and photosynthesis (see refs. [15,76,77]). TCA cycle tricarboxylic acid cycle, ATP adenosine triphosphate, ADP adenosine diphosphate, Pi phosphate ions, OEC oxygen-evolving complex, PQ plastoquinone, PTOX2 the plastid terminal oxidase 2, NDA2 the type II NADPH dehydrogenase, PGR5 proton gradient regulation 5, Cytb6f cytochrome b6f, PC plastocyanin, Fd ferredoxin, NADPH nicotinamide adenine dinucleotide phosphate, FNR ferredoxin-NADP⁺ oxidoreductase, SOD superoxide dismutase, APX ascorbate peroxidase, OAA oxaloacetate, Glu glutamate, LCIA limiting $CO_2$ inducible A, LCI11 low-$CO_2$-inducible membrane protein, DiT2.1 dicarboxylate transporter 2.1, LCI20 low-carbon-inducible 20, CIA8 involved in inorganic carbon acclimation.

by Dual-PAM 100. The Y(I) was determined by Y(I) = Pm′/Pm. Pm and Pm′ are the maximal absorbance changes induced by applying the saturating light under dark and actinic light, respectively[70].

For *P. tricornutum*, *Porphyridium purpureum*, *Chlamydomonas reinhardtii*, *C. pyrenoidosa* and *P. patens*, cells were washed with their fresh medium, respectively. 4.5 mM HAc was added to decrease the pH to 5.5 and the KOH was used to adjust the pH back to 7.5. When NaAc was used to induce fluorescence quenching, final concentration of NaAc of 50 mM was reached. In total, 100 μM nigericin was used to collapse the pH gradient formed across thylakoid membrane.

The eosinophilic algae NIES-2499 was grown in photoautotrophic medium at pH 3.5. When 4.5 mM HAc was added and the pH of medium remained basically unchanged.

Chlorophyll fluorescence images of *Arabidopsis thaliana, Triticum aestivum* and *Sorghum bicolor* were obtained with Imaging-PAM under aerobic conditions. Actinic light of 400 μmol photons m⁻² s⁻¹, saturating light (4000 μmol photons m⁻² s⁻¹, 800 ms) and a red measuring light was used in this experiment. Before measurements, all plants were dark-incubated for 30 min. For *Arabidopsis thaliana*, two leaves were injected with acetic acid (0.1 M), whiles the other two leaves were injected with the same amount of water and other leaves were untreated as controls. For *Triticum aestivum* and *Sorghum bicolor*, two

different positions of one leaf were injected with acetic acid (0.1 M), and the other two locations were injected with the water as control.

## Oxygen measurements

Oxygen concentration in solution and chlorophyll fluorescence were simultaneously recorded at 24 °C with a commercial oxygen electrode (the FireSting-$O_2$, a PC-controlled (USB) fiber-optic oxygen meter was purchased from the PyroScience) and Dual-PAM 100. To maintain an anaerobic condition, cells of *Chlamydomonas* were typically concentrated to chlorophyll concentration of 40 μg/ml and kept in an airtight cuvette with HSM (*C. reinhardtii*) or M-Allen medium (NIES-2499) mixed with ficoll (10%). The Fm′ in the dark was recorded with a saturating light at an interval of 3 min. When the fluorescence quenching became stable, KOH was added to medium (pH lifted from 6.4 to 7.4) to recover the fluorescence.

## Light-response curves of photosynthesis

The electron transport rates (ETRs) was calculated as ETR (I) or ETR(II) = 0.5 × 0.84 × PAR × Y(I) or Y(II), where 0.5 is the fraction of absorbed light reaching PS I or PSII and 0.84 is absorbed irradiance taken as 0.84 of incident irradiance. For all ETR measurements, actinic light (AL) was turned on 30 s before a saturation pulse

(6000 µmol photons m$^{-2}$ s$^{-1}$, 250 ms pulse duration) was applied. And a series of actinic light with intensity ranging from 0, 13, 54, 89, 167, 267, 416, 754, 1173, 1450 µmol photons m$^{-2}$ s$^{-1}$ was sequentially measured (Fig. 6a, b and Supplementary Fig. 11a).

## Determination of cytosolic pH

Before imaging, cell wall-free WT *Chlamydomonas* CC-400 was resuspended to a concentration of $2 \times 10^7$ cells/ml in NMG buffer (10 mM HEPES, 60 mM KCl, 3 mM MgCl$_2$ pH 6.8). The cells were incubated at 36 °C for 1 h in the NMG buffer containing 5 µM BCECF-AM[71]. The BCECF-AM could diffuse through the cell membrane and intracellular esterase cleave the ester bond releasing BCECF[72]. Then, the dye-stained CC-400 cells were washed twice with fresh NMG buffer to remove the extracellular dyes. For confocal imaging experiments (Leica TCS SP5), excitation wavelength of 488 nm was used and fluorescence emission at 530 ± 15 nm was collected.

BCECF fluorescence is pH sensitive, of which the ratio of 490/440 nm was used to indicate the pH values[73]. The titration curve of pH intracellular in CC-400 loaded with the BCECF-AM was performed by using a FLS1000 Photoluminescence Spectrometer (Edinburgh) with an emission wavelength at 530 nm and the ratio excitation wavelengths of 490 and 439 nm[72]. Slit widths were set to 3 nm. Each sample was placed under different pH buffer and 100 µM nigericin was added to ensure an equilibrium in and out of the cell at external pH values that ranging from 4.0 to 9.0. The pH values were adjusted by using 1 M KOH or 1 M HCl. The BCECF-stained cells in aerobic or anaerobic conditions (darkness for 3 h and sealed for anaerobic respiration) were treatment with acetic acid or NaAc to evaluate the cytosolic pH.

## Organic acid analysis

Organic acid analysis was performed by liquid chromatography (HPLC; Waters 550, Waters, MA, USA) using a Hypsil C18 column (5 µm, 4.6 mm × 250 mm). The mobile phase was a solution containing 0.02 M KH$_2$PO4 at pH 2.4 with the orthophosphoric acid. Flow rate was 0.6 mL/min and the detection wavelength was at 210 nm. We measured the main weak acids produced by fermentation—formate and acetate. Anaerobically adapted cells were collected at the interval of 30 min and then, centrifugation at 9568 × *g* for 2 min to be measured. The samples were filtered with a 0.22 µm aqueous filter. Then, 50 µl of sample was injected onto the column. Retention peaks were recorded using Agilent Chem Station software, and quantification was performed by comparisons with the standard curve for weak acid content.

## Electrochromic shift analysis

The ECS signal was measured by the DUAL-PAM 100 with the P515 module, see details in ref. 74. The cells *npq4 stt7-9* were resuspended with fresh HSM medium at 20 µg/mL chlorophyll. For anaerobic condition, the cells were sealed and placed in darkness for 3 h. Before each measurement, cells were dark-adapted for 30 min. The changes of ECS signal was induced by a single turnover flash (ST, 50 µs). The kinetics of this signal contain three phases[75]: the fast phase-a represents the charge separation of PSI and PSII, then a slower rising phase-b shows the cytochrome b6f activity. and the last phase-c represents the activity of the CF0-F1 ATP synthase.

## Reporting summary

Further information on research design is available in the Nature Portfolio Reporting Summary linked to this article.

## Data availability

All data are available in the main text or the Supplementary Materials. The data that support the findings of this study are available from the corresponding author upon request. Source data are provided with this paper.

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

## Acknowledgements

This work was supported by grants from the National Key R&D Program of China (No. 2019YFA0904600), the National Natural Science Foundation of China (Grant Nos. 11804172, 31970381), the Strategic Priority Research Program of the Chinese Academy of Sciences, Grant No. XDA 26030201 and the Dutch Organization for Scientific research (NWO) via a Vici grant.

## Author contributions

L.T., X.P., W.J.N. and R.C. designed the experiments. X.P., M.Z., J.J., Y.F. and W.Y. performed research; P.C. did the fluorescence experiments on the icl mutant. X.P. analyzed data; all authors discussed the results and commented on the manuscript. L.T. and X.P. wrote the manuscript. L.T., X.P., W.J.N. and R.C. modified the article.

## Competing interests

The authors declare no competing interests.
