## [Peer Review File · Nature Communications]

Weak acids produced during anaerobic respiration suppress both photosynthesis and aerobic respirationReviewer #1 (Remarks to the Author):

Chlamydomonas reinhardtii is a metabolically flexible green microalgae with phototrophic, heterotrophic, and fermentation metabolic capacities. The activation of fermentation occurs under microoxic or anoxic conditions, typically dominating under dim light or darkness, and allows the cells to sustain energy production in the absence of respiration, while producing various weak acids, such as pyruvate, lactate, formic acid, acetic acid. Under long-term dark adaptation, cells experience anoxia and thylakoid luminal acidification occurs, proposed to depend on either chlororespiration or ATP hydrolysis in darkness. In a previous work, the authors have demonstrated that addition of weak acids into a suspension of *Chlamydomonas* cells induces LHCSR-dependent non-photochemical quenching (NPQ; i.e. qE) of chlorophyll fluorescence. In the present manuscript the authors hypothesized that the thylakoid luminal acidification in the dark is caused by the generation of fermentative metabolites and they studied the impact of those fermentative products on photosynthesis, which has not been studied so far.

By measuring chlorophyll fluorescence of *Chlamydomonas* during fermentation in the *stt7* mutant, unable to perform state transitions, they first show that dark anaerobiosis triggers LHCSR-mediated NPQ, and taking advantage of the fact that LHCSR acts as pH sensor they concluded that dark anaerobiosis leads to luminal acidification. Luminal acidification could also be reproduced by exogenously adding weak acids and this acidification (always probed as NPQ in *stt7*) was rapidly reversed once the pH was set back to 7 with the addition of KOH. Acetic acid was able to trigger NPQ not only in *Chlamydomonas* but also in other microalgae and higher plants. The role of chlororespiration and ATP hydrolysis in the acetate-induced NPQ in the dark was addressed by using appropriate mutants of *Chlamydomonas*. The authors conclude that neither of the two processes are involved in the acetate-induced NPQ in the dark and they rather favored the hypothesis of a passive diffusion of the weak acids into the thylakoid lumen resulting in a drop of the pH that activates LHCSR-driven NPQ; this was based on the use of pH-sensitive dyes to estimate the cytoplasmic pH and modeling to predict the thylakoid luminal pH. Finally, the impact of acetic acid and of the pH on photosynthesis and respiration was evaluated; the authors found that both aerobic respiration and photosynthesis are inhibited by acetate and they propose weak acids play an important role for the transition from dark anoxia to low light for the faster establishment of aerobiosis.

Overall, the manuscript advances our understanding on the physiological role of fermentative metabolism and I find the proposed link between fermentative metabolism and regulation of photosynthesis intriguing. At the same time though, the evidence presented here is sometimes suggestive and important controls that would consolidate the conclusions are missing. Additionally, clarity needs to be improved; several important details on experimental setups and the conclusions are either missing or very briefly presented, making it difficult for the reader (especially for a non-specialist) to follow the storyline. Please find below my specific comments.

Major comments

1. The accumulation of fermentative products during the experimental conditions used, e.g. Fig. 1, is only assumed based on published reports. It will be essential that the fermentative metabolites are measured in order to strengthen the claimed link between accumulation of weak acids and activation of LHCSR quenching. For instance, we can see a clear drop of F_m' even at the third pulse (6 min, assuming that the first pulse is at $t=0$). Is there already an accumulation of weak acids after 6 min even though O_2 is still present? What is the profile of the metabolites and what are their concentrations? This information will be valuable to make a direct link with the next part of the work where weak acids are added exogenously but also to compare with the existing literature, especially because different WT strains are used.

2. About the different WT strains used in the study; please provide more information on the genetic background of each one the mutants used. Given the marked phenotypic variation among the different laboratory WT strains (see Gallaher et al. *Plant Cell* 2015, 27, 2335) I was surprised not to see a comparison of the different WT strains used here in terms of the observed phenotypes (NPQ in the dark, accumulation of fermentative products). This becomes more evident in Fig. 4 where the *pox2* and *nda2* mutants were not compared to their genetic background (I guess the

CC-4533?) and therefore the conclusions of the involvement or not of chlororespiration / ATP hydrolysis in the observed phenotypes is questionable. The authors exclude a role of those two processes based on the fact that the dark quenching does occur in the two mutants when NaAc is added and this is reversed by the addition of nigericin. However, no WT control is included. From the raw data of Fig. 2a, a NPQ of ca. 4 can be calculated in WT (I guess CC-124?) treated with HAc, which is much higher than the NPQ of 1 (Fig. 4c) in *ptox2* and *nda2*. Is this difference due to the mutations or due to the different genetic backgrounds of wt and mutants? This needs to be addressed.

3. Fig. 2: why did the authors switch from HAc to NaAc between Fig. a and b? What happens if nigericin is added in the presence of HAc? What is the rationale of choosing to use the very high concentration of 50 mM NaAc?

4. Showing LHCSR3/1 protein levels in the cells used at each of the experiment will be important given the key role of these proteins as pH sensors inferring the pH of the thylakoid lumen. Also, the culture conditions in each Figure are not properly mentioned; e.g. Fig. 1: how long were the cells exposed to HL before the experiment in the dark; Fig. 4: what were the culture conditions before the fluorescence measurements?

5. Does acetate get partly metabolized immediately after its exogenous addition and uptake? If yes how much of this plays a role on the observed phenotypes? The *icl* mutant, deficient in isocitrate lyase and with severe defect in acetate metabolism, will be instrumental to address the question of the participation of acetate metabolism in the phenotypes.

6. Where are weak acids produced within the cells? Are the acids simply diffusing through the membranes reaching either the extracellular medium and the chloroplast thylakoid lumen? Acetate is produced both in the mitochondria and in the chloroplast under fermentative conditions. How does the model of the author integrate this important aspect? According to the model how much of the acid is needed to trigger pH changes? Have the authors tried to do titration experiments with concomitant measurements of the uptake to get a better grasp of this phenomenon?

7. How was the pH regulated in Fig. 6d? Was it by adding different amounts of acetate?

8. The conclusions on the impact of weak acids on aerobic respirations are not very clear to me. Following the logic of the manuscript and according to Fig. 2b, NaAc triggers NPQ in the dark and this was attributed to the luminal acidification. If weak acids also inhibit aerobic respiration, as proposed in lines 197-198, one should also expect that addition of NaAc would inhibit respiration. To me this counter-intuitive because it is well documented that adding sodium acetate in HSM cells boosts respiration. Can the authors elaborate on this?

9. The genotyping of CLiP mutants often represents a challenge. In the case of the *ptox* (intronic mutant) and *nda2* (5'UTR insertion) validation of only one junction insertion site is not sufficient to prove that the strains are indeed mutants. RT-PCR or RT-qPCR should be employed to show that PTOX2, NDA2 gene expression is abolished in the mutants. Finally, it would be much more appropriate to use the correct wt for the two mutants.

Minor points

- Fig. 1: Why is KOH added to the medium? Although explained in M&M it should also be added at the first instance in the main text.

- Lines 107-108: A list of weak acids inducing this effect is shown in Supplementary Fig. 2 and Supplementary Table. 1 : add the concentrations that were used.

- Fig. 3. What is the biological relevance of using diatoms, Arabidopsis etc.? Do these organisms have fermentative metabolism? Is the scope of this experiments simply to demonstrate that exogenous added acetate can trigger luminal acidification and NPQ? Also how can one explain the very slow kinetics of recovery after adding KOH in the Pt?

- How does KOH impact luminal pH? Is it a passive transfer?
- Luminal and Lumenal is used throughout the text.
- Fig. 4d is not described in the main text in sufficient detail; e.g. the reason why DCMU was added is only briefly explained in the figure legend.
- Fig. 6c: Are the measurements performed on equal cell numbers or on equal chlorophyll? Showing the raw data (micromol O₂/cell or chl) will allow for comparisons with other studies.

Reviewer #2 (Remarks to the Author):

In this paper, Pang and co-workers study the effect of weak acidic compounds on the luminal pH by following the qE component of the chlorophyll fluorescence non-photochemical quenching. Indeed, qE has been previously shown to be related to lumen acidification due to the reversible protonation of the LHCSR3 and LHCSR1 proteins. Using a similar approach, Tian et al. (2019) previously reported that an exogenous supply of acetic acid to *Chlamydomonas* cells can trigger lumen acidification, the effect being reversed by KOH addition. In this paper, Pang et al. extend this work by showing a similar using other weak organic acids (malic, succinic, lactic and citric acid...). By using a state transition mutant (stt7) and a double mutant npq4 stt7 the effects of weak acid supply are clearly attributed to changes in qE. Most of the work is conducted on the green algal *Chlamydomonas* a species that is able of both photosynthesis and fermentation metabolisms and can switch from on to the other depending on the growth conditions. Based on the effect of acidic compounds on qE, the authors extend to other algal species and to land plants.

By comparing the effect of exogenously supplied weak acids to the effect of anaerobiosis, which is supposed from the literature to trigger the production of organic acids by fermentation, the authors conclude that the production of acidic compounds during fermentation acidifies the lumen and affects photosynthesis. However, this conclusion is not supported by experimental data:

- In order to convincingly conclude that weak acids produced by fermentation are indeed involved in observed NPQ changes, the authors should measure the intracellular levels of fermentation products.
- The authors use the *Chlamydomonas eustigma* strain NIES-2499, which they argue lacks enzymes involved in organic acid fermentation and show the absence of qE change in response to anaerobiosis in this strain. However, no reference to previous work is supplied to justify the choice of this strain. And again, in the absence of quantification of intracellular amounts of organic acids comparing *C. reinhardtii* and *C. eustigma* it is difficult to be convinced that weak acids produced by fermentation are indeed responsible for differences in the luminal pH observed between these species.

Then, the authors use chlororespiration mutants (ptox2 and nda2) to exclude a contribution of chlororespiration to changes in the luminal pH in the dark. However, the mutants used for that purpose have not been sufficiently characterized. It is referred to the publication of Desplats et al. (2009) to justify the choice of the strain (LMJ.RY0402.206160), which is rather surprising since this paper did not mention the use of any nda2 mutant and since the CLiP library was made available in 2016. To my knowledge, the only Nda2 deficient strain characterized so far was published by Jans et al. (2008) and the authors generated the mutant strain by RNA interference. In order to qualify the CLiP mutant LMJ.RY0402.206160 as a NDA2 knock-out mutant the authors

should not only rely on the verification of the insertion site, but should show the absence of transcript and protein. This is particularly critical since according to the CLiP website, the LMJ.RY0402.206160 holds an insertion in the 5'UTR. The same for the ptox2 mutant for which the authors refer to the work of Houille-Vernes et al. (2011) but use a different strain (the CLiP mutant strain LMJ.RY0402.152174 which harbors an insertion in an intron).

Finally, the authors study the effect of exogenously supplied weak acid on the photosynthetic electron transport activity and report a quite strong effect which is interpreted by a down-regulation of the cytb6f complex. However, if the effect of weak acid supply is quite clear, the effect of anaerobic fermentation on photosynthesis has been only scarcely characterized (Supplemental Figure 9). The authors should more carefully study the effects of fermentation products on photosynthesis by measuring intracellular amounts of fermentation products in different conditions of fermentation and correlate these effects with measurements of photosynthesis activity. Therefore, the title seems partly not justified by the experimental data ("anaerobic respiration suppresses photosynthesis") and partly enigmatic ("anaerobic respiration suppresses aerobic respiration") as it seems obvious that in the absence of oxygen aerobic respiration will be suppressed).

All together this paper reports interesting data related to the effects of exogenously supplied weak organic acids on the luminal pH acidification analyzed by NPQ measurements. However, the link with the fermentation metabolism and the internal production of organic acids is not sufficiently robustly established and the claim that "weak acids produced during fermentation down-regulate photosynthesis" is not sufficiently justified by the experimental data.

Answers to the reviewers' comments point by point:

Reviewer 1:

Chlamydomonas reinhardtii is a metabolically flexible green microalgae with phototrophic, heterotrophic, and fermentation metabolic capacities. The activation of fermentation occurs under microoxic or anoxic conditions, typically dominating under dim light or darkness, and allows the cells to sustain energy production in the absence of respiration, while producing various weak acids, such as pyruvate, lactate, formic acid, acetic acid. Under long-term dark adaptation, cells experience anoxia and thylakoid luminal acidification occurs, proposed to depend on either chlororespiration or ATP hydrolysis in darkness. In a previous work, the authors have demonstrated that addition of weak acids into a suspension of *Chlamydomonas* cells induces LHCSR-dependent non-photochemical quenching (NPQ; i.e. qE) of chlorophyll fluorescence. In the present manuscript the authors hypothesized that the thylakoid luminal acidification in the dark is caused by the generation of fermentative metabolites and they studied the impact of those fermentative products on photosynthesis, which has not been studied so far.

By measuring chlorophyll fluorescence of *Chlamydomonas* during fermentation in the *stt7* mutant, unable to perform state transitions, they first show that dark anaerobiosis triggers LHCSR-mediated NPQ, and taking advantage of the fact that LHCSR acts as pH sensor they concluded that dark anaerobiosis leads to luminal acidification. Luminal acidification could also be reproduced by exogenously adding weak acids and this acidification (always probed as NPQ in *stt7*) was rapidly reversed once the pH was set back to 7 with the addition of KOH. Acetic acid was able to trigger NPQ not only in *Chlamydomonas* but also in other microalgae and higher plants. The role of chlororespiration and ATP hydrolysis in the acetate-induced NPQ in the dark was addressed by using appropriate mutants of *Chlamydomonas*. The authors conclude that neither of the two processes are involved in the acetate-induced NPQ in the dark and they rather favored the hypothesis of a passive diffusion of the weak acids into the thylakoid lumen resulting in a drop of the pH that activates LHCSR-driven NPQ; this

was based on the use of pH-sensitive dyes to estimate the cytoplasmic pH and modeling to predict the thylakoid luminal pH. Finally, the impact of acetic acid and of the pH on photosynthesis and respiration was evaluated; the authors found that both aerobic respiration and photosynthesis are inhibited by acetate and they propose weak acids play an important role for the transition from dark anoxia to low light for the faster establishment of aerobiosis.

Overall, the manuscript advances our understanding on the physiological role of fermentative metabolism and I find the proposed link between fermentative metabolism and regulation of photosynthesis intriguing. At the same time though, the evidence presented here is sometimes suggestive and important controls that would consolidate the conclusions are missing. Additionally, clarity needs to be improved; several important details on experimental setups and the conclusions are either missing or very briefly presented, making it difficult for the reader (especially for a non-specialist) to follow the storyline. Please find below my specific comments.

We are glad to hear that the reviewer finds our work novel and intriguing and we would like to thank them for the help in strengthening the conclusions. Based on their suggestions, we have now performed a series of additional experiments that are discussed in detail below.

Major Comments:

1. The accumulation of fermentative products during the experimental conditions used, e.g. Fig. 1, is only assumed based on published reports. It will be essential that the fermentative metabolites are measured in order to strengthen the claimed link between accumulation of weak acids and activation of LHCSR quenching.

We have now measured the fermentative metabolites at various time. The data correlate very well the NPQ level. These data are added to Fig. 1 (panels e, f).

Fig.1: The relationship between fermentation products formic acid (e) and acetic acid (f) accumulated in the medium of *Chlamydomonas* cultures and NPQ in *stt7-9*. Cells were in the dark anaerobiosis for up to 4.5 h. Samples were taken at the indicated time points (0, 0.5, 1, 1.5, 2, 2.5, 3, 4 and 4.5 h), centrifuged, and filtered, and the medium was analyzed by HPLC. Data are taken from triplicate samples derived from three independent experiments. Error bars represent S.D.

2. “For instance, we can see a clear drop of Fm’ even at the third pulse (6 min, assuming that the first pulse is at $t=0$). Is there already an accumulation of weak acids after 6 min even though O_2 is still present?”

The O_2 was very rapidly depleted (~ minutes timescale; see the oxygen consumption in the lower panel of Fig. 1a). After 6 minutes, indeed, there is a clear drop of Fm’, however, it is not significant as NPQ value at this point is only ~0.22, and its maximum NPQ is ~3.5, which suggests that the lumen pH is only marginally reduced.

3. “What is the profile of the metabolites and what are their concentrations?”

The final concentration of formic acid and acetic acid in the culture supernatant after ~3h fermentation was 2.77 mM and 1.3 mM, respectively. These results are in line with the previous results obtained by Mus et al., *J Biol Chem* 2007.

4. About the different WT strains used in the study; please provide more information on the genetic background of each one the mutants used.

This information is now added in Supplementary Figure 8. According the ref Li et al., *Plant Cell* 2016, the background strain of *ptox2* and *nda2* is CC-4533.

5. Given the marked phenotypic variation among the different laboratory WT strains (see Gallaher et al. *Plant Cell* 2015, 27, 2335) I was surprised not to see a comparison

of the different WT strains used here in terms of the observed phenotypes (NPQ in the dark, accumulation of fermentative products). This becomes more evident in Fig. 4 where the *ptox2* and *nda2* mutants were not compared to their genetic background (I guess the CC-4533?) and therefore the conclusions of the involvement or not of chlororespiration / ATP hydrolysis in the observed phenotypes is questionable. The authors exclude a role of those two processes based on the fact that the dark quenching does occur in the two mutants when NaAc is added and this is reversed by the addition of nigericin. However, no WT control is included. From the raw data of Fig. 2 a, a NPQ of ca. 4 can be calculated in WT (I guess CC-124?) treated with HAc, which is much higher than the NPQ of 1 (Fig. 4c) in *ptox2* and *nda2*. Is this difference due to the mutations or due to the different genetic backgrounds of wt and mutants? This needs to be addressed.

We have now added experiments on the WT strain CC-4533 (Supplementary Figure 7), which is the background strain of *ptox2* and *nda2*. We observed that the NPQ of the highlight adapted CC-4533 is almost 4, which is similar to the *stt7-9* shown in Fig 2.

As the Reviewer pointed out, the NPQ value of Chlamy can be very different in WTs with different genetic backgrounds. However, the large variations in NPQ amplitude is not an issue in our experiments: as long as NPQ can be activated by the weak acid addition, we can use relative values of NPQ to quantify the responses. We have, nonetheless, performed the experiments on WT-CC-4533, the same phenomenon as chlororespiration mutants used in the manuscript.

Supplementary Figure 7. Typical chlorophyll fluorescence traces of CC-4533 measured with Dual-PAM. Cells were illuminated with strong actinic light ($1500 \mu\text{mol photons}\cdot\text{m}^{-2}\cdot\text{s}^{-1}$) for 150 s followed by dark recovery (4 min). The addition of 50 mM NaAc leads to fluorescence quenching (350 s - 590 s) and nigericin (100 μM) releases the quenching (650 s - 850 s).

6. Fig. 2: why did the authors switch from HAc to NaAc between Fig. a and b? What happens if nigericin is added in the presence of HAc?

The motivation behind transitioning from acid- to salt-induced quenching is that the latter permits us to check the “ion trap” hypothesis and highlight the importance of lumen properties in the process. The addition of nigericin has no effect on the acid-induced quenching, as all compartments are equally acidified in HAc presence. We have now added these data (see the figure below).

Figure: Typical chlorophyll fluorescence traces of *stt7-9* measured with Dual-PAM. Cells were illuminated with strong actinic light ($1500 \mu\text{mol photons}\cdot\text{m}^{-2}\cdot\text{s}^{-1}$) for 150 s followed by dark recovery (4 min). The addition of acetic acid (the pH of medium 5.5) caused fluorescence quenching (350 s - 590 s) and nigericin ($100 \mu\text{M}$) couldn't release the fluorescence quenching (650 s - 850 s)

What is the rationale of choosing to use the very high concentration of 50 mM NaAc?

The high salt concentration is required to reach the maximal amplitude of quenching. Upon addition of NaAc, the pH of the culture solution remains neutral, this pH would shift the equilibrium towards acid dissociation, namely, high $[\text{H}^+]$ and $[\text{Ac}^-]$, but low $[\text{HAc}]$. Only with a high concentration of NaAc, $[\text{HAc}]$ becomes high enough to induce the maximum quenching.

7. Showing LHCSR3/1 protein levels in the cells used at each of the experiment will be important given the key role of these proteins as pH sensors inferring the pH of the thylakoid lumen. Also, the culture conditions in each Figure are not properly mentioned; e.g. Fig. 1: how long were the cells exposed to HL before the experiment in the dark; Fig. 4: what were the culture conditions before the fluorescence measurements?

We have grown the cells under continuous moderate-strong light ($350 \mu\text{mol photons m}^{-2} \text{ s}^{-1}$), and dark-adapted for 20 min prior to experiments, for all the experiments. We have now amended the Methods section. The protein levels of LHCSR3/1 in the same strain (*stt7-9*) grown in the same conditions were reported previously (L. Tian, et al., PNAS 2019). Moreover, here we only use NPQ as an indicator of changes in the thylakoid lumen, which does not require knowing the exact level of the proteins.

8. Does acetate get partly metabolized immediately after its exogenous addition and uptake? If yes how much of this plays a role on the observed phenotypes? The *icl* mutant, deficient in isocitrate lyase and with severe defect in acetate metabolism, will be instrumental to address the question of the participation of acetate metabolism in the phenotypes.

Acetic acid can be metabolized by *Chlamydomonas*, but this process occurs on the hour scales (R. D. Levine, et al., Cells 2019). Moreover, we have tested a series of weak acids, which can not be assimilated by *Chlamydomonas*, and we show that they can all

acidify the lumen.

We have also followed the suggestion of the reviewer and used the *icl* mutant. We found that the quenching effect induced by NaAc was the same as in the reference strain. This result is now added to the manuscript (Supplementary Figure 4).

Supplementary Figure 4. Fm of *icl* mutant and its parental strain. The black bars are the Fms of WT and *icl* mutants at pH 7.2, and the red ones are the Fms after adding 5 mM NaAc at pH 5.5.

9. Where are weak acids produced within the cells? Are the acids simply diffusing through the membranes reaching either the extracellular medium and the chloroplast thylakoid lumen? Acetate is produced both in the mitochondria and in the chloroplast under fermentative conditions. How does the model of the author integrate this important aspect?

According to F. Mus, et al., J Biol Chem 2007, acetate is produced in cytoplasm, mitochondria and chloroplast during fermentation. It is known that weak acids in their neutral form could passively diffuse through membranes. Based on this property of the membrane, we argue that the exact location of weak acids production is not an essential factor, especially given the fact that the externally added acids also induce quenching.

These points are now explained in the introduction and discussion.

10. According to the model how much of the acid is needed to trigger pH changes? Have the authors tried to do titration experiments with concomitant measurements of the uptake to get a better grasp of this phenomenon?

The titration is shown in our previous work (L. Tian, et al., PNAS 2019): NPQ started immediately after adding acetic acid and reached a maximum NPQ when ~ 4.5 mM acetic acid are added to our system.

11. How was the pH regulated in Fig. 6d? Was it by adding different amounts of acetate?

Yes, we added different amounts of acetic acid to adjust the pH to the targeted values. This is now explained in the Fig.6d legend.

12. The conclusions on the impact of weak acids on aerobic respirations are not very clear to me. Following the logic of the manuscript and according to Fig. 2b, NaAc triggers NPQ in the dark and this was attributed to the lumenal acidification. If weak acids also inhibit aerobic respiration, as proposed in lines 197-198, one should also expect that addition of NaAc would inhibit respiration. To me this counter-intuitive because it is well documented that adding sodium acetate in HSM cells boosts respiration. Can the authors elaborate on this?

In the longer term, indeed, aerobic respiration is boosted by adding sodium acetate, but in the short-term, there is an effect called “Substrate inhibition”, especially for acetate (F. Chen, et al., Process Biochemistry 1994). To show this effect, we have monitored how the oxygen consumption rate was affected by acetate. It turned out that the oxygen consumption rate became much slower upon adding 50 mM NaAc, see Supplementary Figure 15 in below.

Supplementary Fig. 15: Oxygen consumption in the dark of cells in aerobic condition in the absence (black line) and presence (red line) of 50 mM NaAc.

13. The genotyping of CLiP mutants often represents a challenge. In the case of the *ptox* (intronic mutant) and *nda2* (5UTR insertion) validation of only one junction insertion site is not sufficient to prove that the strains are indeed mutants. RT-PCR or RT-qPCR should be employed to show that *PTOX2*, *NDA2* gene expression is abolished in the mutants. Finally, it would be much more appropriate to use the correct wt for the two mutants.

RT-PCR were performed on these two mutants confirming the mutations. The results are reported in the Supp. Fig. 16 (in below). The corresponding reference strain WT has also been measured.

Supplementary Figure 16. Identification of the *ptox2*, *nda2* and *FUD50* mutants. **a** Genome DNA was extracted from CC4533 and *ptox2*. PCR with the P-F/C1 primer shows an insertion within the PTOX2 gene in the *ptox2* mutants. **b** Genome DNA was extracted from CC4533 and *nda2*. PCR with the of N-F/C1 and N-R/C2 primers shows an insertion within the NDA2 gene in the *nda2* mutant. **c** PCR amplification of DNA fragment of 500 bp (*atpB* gene) in wild type CC-124 and mutant of *FUD50*. **d** Mapping of the insertion of cassette CIB1 in the nuclear genome of *ptox2* (top) and *nda2* (bottom). The cassette CIB1 insertion site is indicated by black lines. **e** Characterization of transcription of PTOX2 in the CC-4533 and *ptox2* mutant. **f** Characterization of transcription of NDA2 in the CC-4533 and *nda2* mutant. Products obtained from the WT CC-4533, *ptox2* and *nda2* strains by semi-quantitative RT-PCR are shown. Transcript level of CBLP gene was used as a loading control. The correct PCR fragments were also confirmed by DNA sequencing. All primers used are listed in Supplementary Table 4.

14. Minor points

Fig. 1: Why is KOH added to the medium? Although explained in M&M it should also be added at the first instance in the main text.

We have it now added the explanation to the main text (see the lines 96-97).

Lines 107-108: A list of weak acids inducing this effect is shown in Supplementary Fig. 2 and Supplementary Table. 1: add the concentrations that were used.

We have now indicated the concentration of these weak acids in the notes of the chart.

Fig. 3. What is the biological relevance of using diatoms, Arabidopsis etc.? Do these organisms have fermentative metabolism? Is the scope of this experiments simply to demonstrate that exogenous added acetate can trigger luminal acidification and NPQ? Also how can one explain the very slow kinetics of recovery after adding KOH in the Pt?

Using a range of species and their mutants (e.g., *icl*), we wanted to ensure that the “ion trapping” model is a general property of bioenergetic membranes. We do not claim that they possess fermentative metabolism, although in case of non-axenic conditions in nature, photosynthetic microbes can still be impacted by weak acids.

How does KOH impact luminal pH? Is it a passive transfer?

It is due to the chemical equilibrium: by neutralizing the pH outside of the cells, less dissociation takes place in all compartments of the cells.

Luminal and Lumenal is used throughout the text.

Thank you, we have made the corresponding revisions.

Fig. 4d is not described in the main text in sufficient detail; e.g. the reason why DCMU was added is only briefly explained in the figure legend.

Thank you, now we have elaborated on this point in the main text (see the lines 156-157)

Fig. 6c: Are the measurements performed on equal cell numbers or on equal chlorophyll? Showing the raw data (micromol O₂/cell or chl) will allow for comparisons with other studies.

The measurements were performed on equal chlorophyll unless mentioned (see explanations in line 349)

Reviewer 2:

In this paper, Pang and co-workers study the effect of weak acidic compounds on the luminal pH by following the qE component of the chlorophyll fluorescence non-photochemical quenching. Indeed, qE has been previously shown to be related to lumen acidification due to the reversible protonation of the LHCSR3 and LHCSR1 proteins. Using a similar approach, Tian et al. (2019) previously reported that an exogenous supply of acetic acid to *Chlamydomonas* cells can trigger lumen acidification, the effect being reversed by KOH addition. In this paper, Pang et al. extend this work by showing a similar using other weak organic acids (malic, succinic, lactic and citric acid...). By using a state transition mutant (stt7) and a double mutant npq4 stt7 the effects of weak acid supply are clearly attributed to changes in qE. Most of the work is conducted on the green algal *Chlamydomonas* a species that is able of both photosynthesis and fermentation metabolisms and can switch from on to the other depending on the growth conditions. Based on the effect of acidic compounds on qE, the authors extend to other algal species and to land plants.

By comparing the effect of exogenously supplied weak acids to the effect of anaerobiosis, which is supposed from the literature to trigger the production of organic acids by fermentation, the authors conclude that the production of acidic compounds during fermentation acidifies the lumen and affects photosynthesis. However, this conclusion is not supported by experimental data: In order to convincingly conclude that weak acids produced by fermentation are indeed involved in observed NPQ changes, the authors should measure the intracellular levels of fermentation products.

We have measured the concentration of two fermentation products, i.e formic and acetic acids, that were excreted from the cell. The results show that the concentration of weak acids originating from fermentation positively correlates with the level of NPQ. The data are now added to the manuscript (see figure below and text, lines109-117). To summarize, acidic fermentation induces NPQ and its weak acids production correlates with NPQ, while alcoholic fermentation failed in triggering NPQ. Moreover, the exogenously added weak acids can cause NPQ and the ion-trap model has been verified by an in vitro system of liposome by Gabba et al., Biophysical Journal 2019.

Fig1: The relationship between fermentation products formic acid(e) and acetic acid(f) accumulated in the medium of *Chlamydomonas* cultures and npq in *stt7-9*. Cells were in the dark anaerobiosis and then kept for up to 4.5 h. Samples were taken at the indicated time points (0, 0.5, 1, 1.5, 2, 2.5, 3, 4 and 4.5 h), centrifuged, and filtered, and the medium was analyzed by HPLC. Data are taken from triplicate samples derived from three independent experiments. Error bars represent S.D.

The authors use the *Chlamydomonas eustigma* strain NIES-2499, which they argue lacks enzymes involved in organic acid fermentation and show the absence of qE change in response to anaerobiosis in this strain. However, no reference to previous work is supplied to justify the choice of this strain. And again, in the absence of quantification of intracellular amounts of organic acids comparing *C. reinhardtii* and *C. eustigma* it is difficult to be convinced that weak acids produced by fermentation are indeed responsible for differences in the luminal pH observed between these species.

Hirooka et al., PNAS 2017 show that the NIES-2499 strain does not produce formic and acetic acids under fermentation. Under aerobic conditions, acidophilic algae have an acid-induced NPQ capacity (Fig1c). We have now also checked for the presence of formic acid and acetic acid in this alga, but we did not detect them. These results are presented now in Supplementary Figure 2. The results are consistent with previous measurements (S. Hirooka, et al, PNAS 2017).

Supplementary Figure 2. HPLC traces of weak acids extracts from the *stt7-9* and NIES-2499 mutants. The peaks for formic acid (6.5-6.7 min) and acetic acid (9.2-9.5 min) were labeled according to their retention times, respectively. No formic acid or acetic acid was eluted for the *Chlamydomonas eustigma* NIES-2499.

Then, the authors use chlororespiration mutants (*ptox2* and *nda2*) to exclude a contribution of chlororespiration to changes in the luminal pH in the dark. However, the mutants used for that purpose have not been sufficiently characterized. It is referred to the publication of Desplats et al. (2009) to justify the choice of the strain (LMJ.RY0402.206160), which is rather surprising since this paper did not mention the use of any *nda2* mutant and since the CLiP library was made available in 2016. To my knowledge, the only *Nda2* deficient strain characterized so far was published by Jans et al. (2008) and the authors generated the mutant strain by RNA interference. In order to qualify the CLiP mutant LMJ.RY0402.206160 as a *NDA2* knock-out mutant the authors should not only rely on the verification of the insertion site, but should show

the absence of transcript and protein. This is particularly critical since according to the CLiP website, the LMJ.RY0402.206160 holds an insertion in the 5'UTR. The same for the *ptox2* mutant for which the authors refer to the work of Houille-Vernes et al. (2011) but use a different strain (the CLiP mutant strain LMJ.RY0402.152174 which harbors an insertion in an intron).

Our apologies for this confusion, we intended to refer to the literature stating that the two mutants *ptox2* and *nda2* are chlororespiratory mutants. We have now added the analyses of the mutants to the manuscript as Supp.Fig15 (see also answer 13 to reviewer 1)

Finally, the authors study the effect of exogenously supplied weak acid on the photosynthetic electron transport activity and report a quite strong effect which is interpreted by a down-regulation of the *cytb6f* complex. However, if the effect of weak acid supply is quite clear, the effect of anaerobic fermentation on photosynthesis has been only scarcely characterized (Supplemental Figure 9). The authors should more carefully study the effects of fermentation products on photosynthesis by measuring intracellular amounts of fermentation products in different conditions of fermentation and correlate these effects with measurements of photosynthesis activity.

To further demonstrate that fermentation down-regulate photosynthesis, we added new experiments: we measured the *cyt. b_{6f}*-mediated membrane potential changes; and ETR(I) of *npq4 stt7-9* under both anaerobic and aerobic conditions. The results showed that electron transport does slow down upon fermentation. (See the Supplementary Figure 11-12 and the lines 197-199 and 203-208)

Supplementary Figure 11. **a** Light response curve of electron transport rates of PSI (ETR) in *npq4 stt7-9*. Error bar, SD (n=4). **b** Kinetics of the electrochromic signal of *npq4 stt7-9*. The changes of ECS signal was induced by a single turnover flash (ST 50 μ s) after 30 min of dark adaption.

Supplementary Figure 12. Kinetics of the electrochromic signal of the *npq4 stt7-9* cells under aerobic conditions in the absence of chemicals (control cells, black) and in the presence of 4.5 mM HAc (red) or 50 mM NaAc (blue). The changes of ECS signal was induced by a single turnover flash (ST 50 μ s).

Therefore, the title seems partly not justified by the experimental data (“anaerobic respiration suppresses photosynthesis”) and partly enigmatic (“anaerobic respiration suppresses aerobic respiration”) as it seems obvious that in the absence of oxygen aerobic respiration will be suppressed).

We agree, and modified the title into “Weak acids produced in anaerobic respiration suppress both photosynthesis and aerobic respiration”

All together this paper reports interesting data related to the effects of exogenously supplied weak organic acids on the luminal pH acidification analyzed by NPQ measurements. However, the link with the fermentation metabolism and the internal production of organic acids is not sufficiently robustly established and the claim that “weak acids produced during fermentation down-regulate photosynthesis” is not sufficiently justified by the experimental data.

We are very grateful for the Reviewer’s praise and the suggestions. The suggested experiments clearly strengthen the conclusions.

Reviewer #1 (Remarks to the Author):

I appreciate that the authors have performed several new experiments to address my concerns. The addition of this new data has strengthened the conclusions of the manuscript.

There is one major concern that remains, the effect of acidification on respiration. Figure 6d shows that respiration decreases as the buffer becomes more acidic. However, the experimental conditions remain unclear. I copy here point #11 from the document "Responses to reviewers' comments point by point".

- How was the pH regulated in Fig. 6d? Was it by adding different amounts of acetate?
- Yes, we added different amounts of acetic acid to adjust the pH to the targeted values. This is now explained in the Fig.6d legend.

However, Fig.6d legend reads: "Oxygen level measured in stt7-9 cells resuspended in buffers with different pH (as indicated) under aerobic conditions".

Could the authors please clarify this? If different amounts of acetate were indeed added, what was the concentration of acetate used to produce each of the different pH values? Without this information, it is difficult to see the physiological relevance of the finding that the addition of 50 mM NaAc inhibits respiration (Supplementary Fig. 15). Would the intracellular accumulation of weak acids ever reach such high levels to justify the proposition of the substrate inhibition that the authors raise at point #12? Does this inhibition also occur at lower concentrations of NaAc? Finally, I do not agree with the statement (point#8) that acetate metabolism occurs on the hour scale. Once acetate is added to the cell culture, respiration is boosted immediately (time scale of minutes). Two examples demonstrating this are (i) Lewin, 1954, PMID 13221767; a clear boost of respiration is shown 15 min after acetate addition, (ii) Hanawa et al. 2007, PMID 21243526; respiratory CO₂ increases 30 min after acetate addition (see Fig. 6 of the publication).

Minor points

- point #4 from the document "Responses to reviewers' comments point by point": "This information is now added in Supplementary Figure 8. According the ref Li et al., Plant Cell 2016, the background strain of ptox2 and nda2 is CC-4533.". I guess the authors mean Supplementary Fig. 16?
- Figure 6b legend: please specify how much NaAc was added.
- Supplementary Figure 10: less significant points should be used.
- point #8 from the document "Responses to reviewers' comments point by point": There is no publication R. D. Levine, et al., Cells 2019; perhaps the author mean the Cells 2019, 8, 1367; doi:10.3390/cells8111367 ?

Reviewer #2 (Remarks to the Author):

Apart for the last point, the authors have satisfactorily addressed most of my concerns.

The effect of anaerobiosis on ETR is now shown as Supplementary Fig. 11. However, this experiment is far from convincing. In the absence of experimental details, it is difficult to be convinced that the accumulation of weak fermentative acids is responsible for the observed decrease in ETR. First, weak acids accumulation was quantified in this experiment and a correlation between weak acid accumulation and ETR inhibition was not shown as requested in the previous evaluation. Second, since it is well-known that anaerobiosis induces a lag in the induction of photosynthesis, I question whether each light plateau was long enough to ensure that steady state was reached. I assume that the measurements were made sequentially at the different light intensities. The fact that ETR values reach similar levels at high light intensity is surprising and tends to indicate that steady state was indeed not reached in the measurements made at lower

intensities. This should be carefully checked and the raw chlorophyll fluorescence data should be shown.

Minor detail: on fig. 7 the bicarbonate transporter operating at the chloroplast envelope is not CCP1 (which has recently been shown to be targeted to mitochondria) but LCIA. Also, the bicarbonate channel in the thylakoid membrane (CIA8) is now identified as a bestrophin-like, but H⁺ or Na⁺ co-transport has not been described.

Reviewer #1 (Remarks to the Author):

I appreciate that the authors have performed several new experiments to address my concerns. The addition of this new data has strengthened the conclusions of the manuscript.

Thank you very much for your valuable input.

Comment 1: There is one major concern that remains, the effect of acidification on respiration. Figure 6d shows that respiration decreases as the buffer becomes more acidic. However, the experimental conditions remain unclear. I copy here point #11 from the document "Responses to reviewers' comments point by point".

- How was the pH regulated in Fig. 6d? Was it by adding different amounts of acetate?

- Yes, we added different amounts of acetic acid to adjust the pH to the targeted values.

This is now explained in the Fig.6d legend.

However, Fig.6d legend reads: "Oxygen level measured in stt7-9 cells resuspended in buffers with different pH (as indicated) under aerobic conditions".

Could the authors please clarify this? If different amounts of acetate were indeed added, what was the concentration of acetate used to produce each of the different pH values?

- How was the pH regulated in Fig. 6d? Was it by adding different amounts of acetate?

Response: Acetic acid, and not sodium acetate was used in this experiment. The pH was adjusted by sequentially adding different amounts of acetic acid (HAc) into 2.0 mL cell solution. Six points were measured between pH 7.1 to 5.5, sequentially corresponding to concentration of acetic acid of 0 mM, 0.5 mM, 1.5 mM, 2.3 mM, 4.1 mM, and 5.0 mM. Note that, adding NaAc would not reduce the pH of the medium, but only that of the lumen, making the absolute pH calibration impossible. We apologize for not describing the experimental details sufficiently, now we add this information to the legend of Fig. 6, see lines 717-720.

Comment 2: Without this information, it is difficult to see the physiological relevance of the finding that the addition of 50 mM NaAc inhibits respiration (Supplementary Fig. 15). Would the intracellular accumulation of weak acids ever reach such high levels to justify the proposition of the substrate inhibition that the authors raise at point #12? Does this inhibition also occur at lower concentrations of NaAc?

Response: The addition of HAc at concentrations comparable with those measurements when the cells were kept in anaerobiosis in the dark (Fig. 1e, f) inhibit oxygen consumption (Fig. 6d), thus confirming the physiological significance of the influence of fermentation on the rest of metabolism.

The pH of the culture solution remained neutral (~7.0) when acetate salt was added. And at this high pH, most of NaAc stays in its ionized form (Na^+ , Ac^-), thus the concentration of the undissociated form of [HAc] is much lower compared to the case when an equal amount of HAc was added, in which the pH of the solution drops to ~5.5 that favors the formation of [HAc]. Therefore, to acidify the lumen a much higher concentration of NaAc than HAc is needed. The experiment in Supplementary Fig. 15, and in particular the high concentration of the salt, was added as an additional demonstration of the validity of the ion trap model.

We have now clarified this in the legend of Supplementary Fig. 15 by adding the sentence in the following:

“Note that, much higher concentration of NaAc than of HAc is required (Fig. 6 d) to inhibit the aerobic respiration, again validifying the proposed ion trap model, see main text.”

--Does this inhibition also occur at lower concentrations of NaAc?

Lower concentrations of NaAc also cause photosynthesis inhibition, but the effect is lower. See the explanation above.

Comment 3: Finally, I do not agree with the statement (point#8) that acetate metabolism occurs on the hour scale. Once acetate is added to the cell culture, respiration is boosted immediately (time scale of minutes). Two examples demonstrating this are (i) Lewin, 1954, PMID 13221767; a clear boost of respiration is shown 15 min after acetate addition, (ii) Hanawa et al. 2007, PMID 21243526; respiratory CO_2 increases 30 min after acetate addition (see Fig. 6 of the publication).

Response: Indeed, the statement is not correct, thank you for pointing it out. Nevertheless, we have tested several weak acids that cannot be metabolized by *Chlamydomonas*, but are able to induce lumen acidification. In addition, as suggested by the reviewer, we show that the *icl* mutant with a defect in acetate metabolism

expresses the same phenomenon upon adding acetic acid.

Minor points

Comment 4: point #4 from the document "Responses to reviewers' comments point by point": "This information is now added in Supplementary Figure 8. According the ref Li et al., Plant Cell 2016, the background strain of ptox2 and nda2 is CC-4533.". I guess the authors mean Supplementary Fig. 16?

Response: Yes, thank you, corrected.

Comment 5: Figure 6b legend: please specify how much NaAc was added.

Response: Instead of using NaAc, 4.5 mM HAc. was added. And it is now specified in line 711.

Comment 6: Supplementary Figure 10: less significant points should be used.

Response: We thank the reviewer for this suggestion. We have done the variance analysis and show that there is no significant difference between the cells with/without NaAc. See the Supplementary Fig. 10.

Supplementary Figure 10. Estimation of cell size. The diameter of CC-400 was calculated from Fig. S9. The green bars represent the diameter of the cells without any treatment, and the blue bars show the diameter of the cells upon addition of 50 mM NaAc. Each data point represents an average value from three independent measurements. Error bar, SD ($n \geq 90$). "a" denotes the absence of statistically significant difference ($p < 0.05$, with one-way ANOVA test performed).

Comment 7: point #8 from the document "Responses to reviewers' comments point by point": There is no publication R. D. Levine, et al., Cells 2019; perhaps the author mean the Cells 2019, 8, 1367; doi:10.3390/cells8111367 ?

Response: We thank the reviewer for pointing out this mistake. It is now fixed.

Reviewer #2 (Remarks to the Author):

Apart for the last point, the authors have satisfactorily addressed most of my concerns.

Comment 1: The effect of anaerobiosis on ETR is now shown as Supplementary Fig. 11. However, this experiment is far from convincing. In the absence of experimental details, it is difficult to be convinced that the accumulation of weak fermentative acids is responsible for the observed decrease in ETR.

Response: Yes, it is an important experimental detail that is required to evaluate the data. In fact, there is a brief description regarding the ETR measurements in the main text, see material and method, however, Supplementary Fig. 11 was not referred there, and now we have it added, see lines 357-361.

“For all ETR measurements, actinic light (AL) was turned on 30 seconds before a saturation pulse ($6000 \mu\text{mol photons}\cdot\text{m}^{-2}\cdot\text{s}^{-1}$, 250 ms pulse duration) was applied. And a series of actinic light with intensity ranging from 0, 13, 54, 89, 167, 267, 416, 754, 1173, $1450 \mu\text{mol photons}\cdot\text{m}^{-2}\cdot\text{s}^{-1}$ was sequentially measured (Fig. 6a, b, and supplementary Fig. 11a).”

Comment 2: First, weak acids accumulation was quantified in this experiment and a correlation between weak acid accumulation and ETR inhibition was not shown as requested in the previous evaluation.

Response: We thank the reviewer for their concerns. We have decided not to overcomplicate the presentation: it is widely established that NPQ is a process that suppresses ETR(II) (by lowering the quantum yield of PSII). The influence of acid accumulation on the induction of NPQ in current work and previous paper (Tian et al., 2019) is clearly indicative of the acid effect on light harvesting and thus electron transfer. Critically, the ETR suppression effect is also present in the double mutant *npq4 stt7-9*. Now we newly characterized the relationship between ETR(I) and fermentation time, the results show that ETR(I) gradually decreased along with the fermentation process (ETR(I) vs Fermentation time) as presented in supplementary Fig 11b. This indicates that the acidification has an additional effect on electron transfer. While we think the major effect is at the level of cyt. b6f efficiency (i.e. “photosynthetic control”), a range of other influences could be established. These include for example a pH effect on any

of the enzymes involved in the CBB cycle, and disentangling and quantifying these effects goes beyond the scope of our paper. We think that the physiological significance of the acidification is clear to see.

Supplementary Figure 11. a Light-response curve of relative electron transport rate of PSI (ETR(I)) in *npq4 stt7-9* under different actinic light intensities. Error bar, SD (n=3). b Light-responses of relative electron transport rate of PSI (ETR (I)) in *npq4 stt7-9* at different time points (0, 1 h, 2 h, and 3 h) in anaerobiosis, actinic light of $754 \mu\text{mol photons}\cdot\text{m}^{-2}\cdot\text{s}^{-1}$ was used. Error bar, SD (n=3). c Kinetics of the electrochromic signal of *npq4 stt7-9*. The change of ECS signal was induced by a single turnover flash (ST 50 μs).

Comment 3: Second, since it is well-known that anaerobiosis induces a lag in the induction of photosynthesis, I question whether each light plateau was long enough to ensure that steady state was reached. I assume that the measurements were made sequentially at the different light intensities.

Response: We agree that the lag in the activation of photosynthesis is present, and it might very well be the same process described here. In our hands, 30 seconds at each intensity is enough, we have tested the signal for up to 3 min and the ETR(I) is stable, see our response to comment 4.

Regarding how Supplementary Fig. 11 a was measured, the reviewer is correct, the different light conditions were sequentially measured and for each intensity, 30 second illumination was applied. These experimental details are now included in the main text, see lines 358-361.

Comment 4: The fact that ETR values reach similar levels at high light intensity is surprising and tends to indicate that steady state was indeed not reached in the measurements made at lower intensities. This should be carefully checked and the raw chlorophyll fluorescence data should be shown.

Response: Yes, the relative difference becomes smaller as light intensity increases. This is because photosynthesis was gradually activated upon the sequential light illumination (10 intensities were measured in a row with 30 seconds for each intensity). However, by monitoring the development of ETR during actinic light illumination, we found that the ETR values are rather stable for the first 3 minutes before its rising, thanks to the lack of both state transition and NPQ in this double mutant. See the figure below (for review only).

Fig: Light-response curve of relative electron transport rate of PSI (ETR(I)) in *npq4 stt7-9* at actinic light $754 \mu\text{mol photons}\cdot\text{m}^{-2}\cdot\text{s}^{-1}$ illumination for 3 min. The cells were kept in anaerobic conditions for 1 h, 2 h, and 3 h.

The ETR(I) measurements are based on the differential absorption measurements of P700 (a commercial module of DUAL-PAM-100 was used)

Comment 5: Minor detail: on fig. 7 the bicarbonate transporter operating at the chloroplast envelope is not CCP1 (which has recently been shown to be targeted to mitochondria) but LCIA. Also, the bicarbonate channel in the thylakoid membrane (CIA8) is now identified as a bestrophin-like, but H^+ or Na^+ co-transport has not been described.

Response: Thank you very much for sharing your expertise here. We had this figure updated, see Fig 7.

Fig. 7. Simplified scheme of metabolic pathways for *Chlamydomonas* under anoxic conditions. The figure illustrates the possible interplays between fermentation, respiration and photosynthesis (see ref.^{15,73,74}). TCA cycle: Tricarboxylic acid cycle. ATP: Adenosine triphosphate. ADP: Adenosine diphosphate. Pi: phosphate ions. OEC: Oxygen-Evolving Complex. PQ: Plastoquinone. PTOX2: the plastid terminal oxidase 2. NDA2: the type II NADPH dehydrogenase. PGR5: Proton Gradient Regulation 5. Cytb6f: Cytochrome b6f. PC: plastocyanin. Fd: Ferredoxin. NADPH: Nicotinamide adenine dinucleotide phosphate. FNR: Ferredoxin-NADP⁺ oxidoreductase. SOD: Superoxide dismutase. APX: Ascorbate peroxidase. OAA: Oxaloacetate. Glu: Glutamate. LCIA: Limiting CO₂ Inducible A. LCI11: low-CO₂-inducible membrane protein. DiT2.1: Dicarboxylate transporter 2.1. LCI20: low-carbon-inducible 20. CIA8: involved in inorganic carbon acclimation.

Reviewer #1 (Remarks to the Author):

All the comments I made were properly addressed by the authors.

Reviewer #2 (Remarks to the Author):

The authors have satisfactorily addressed my last concerns.